# Pointwise Bounds for Distribution Estimation under Communication Constraints

**Wei-Ning Chen**
Department of Electrical Engineering
Stanford University
wnchen@stanford.edu

**Peter Kairouz**
Google Research
kairouz@google.com

**Ayfer Özgür**
Department of Electrical Engineering
Stanford University
aozgur@stanford.edu

## Abstract

We consider the problem of estimating a $d$-dimensional discrete distribution from its samples observed under a $b$-bit communication constraint. In contrast to most previous results that largely focus on the global minimax error, we study the local behavior of the estimation error and provide *pointwise* bounds that depend on the target distribution $p$. In particular, we show that the $\ell_2$ error decays with $O\left(\max\left(\frac{\|p\|_{1/2}}{n2^b}, \frac{1}{n}\right)\right)$ when $n$ is sufficiently large, hence it is governed by the *half-norm* of $p$ instead of the ambient dimension $d$. For the achievability result, we propose a two-round sequentially interactive estimation scheme that achieves this error rate uniformly over all $p$. This two-round scheme extends to $\ell_q$ loss with $q \geq 1$, and hence gives pointwise upper bounds on $\ell_q$ error. We also develop a new local minimax lower bound with (almost) matching $\ell_2$ error, showing that any interactive scheme must admit a $\Omega\left(\frac{\|p\|_{(1+\delta)/2}}{n2^b}\right) \ell_2$ error for any $\delta > 0$.

Our upper and lower bounds together indicate that the $\mathsf{H}_{1/2}(p) \triangleq \log(\|p\|_{1/2})$ bits of communication is both sufficient and necessary to achieve the optimal (centralized) performance, where $\mathsf{H}_{1/2}(p)$ is the Rényi entropy of order 2. Therefore, under the $\ell_2$ loss, the correct measure of the local communication complexity at $p$ is its Rényi entropy.

## 1 Introduction

Learning a distribution from its samples has been a fundamental task in unsupervised learning dating back to the late nineteenth century [1]. This task, especially under distributed settings, has gained growing popularity in the recent years as data is increasingly generated "at the edge" by countless sensors, smartphones, and other devices. When data is distributed across multiple devices, communication cost and bandwidth often become a bottleneck hampering the training of high-accuracy machine learning models [2]. This is even more so for federated learning and analytics type settings [3] which rely on wireless mobile links for communication.

To resolve this issue, several communication-efficient distribution learning schemes have been recently proposed and studied in the literature (see Section 1.2 for a thorough discussion). On the positive side, the state-of-the-art schemes are known to be worst-case (minimax) optimal as they have been shown to achieve the information-theoretic lower bounds on the global minimax error [4–8]. On the

35th Conference on Neural Information Processing Systems (NeurIPS 2021).

negative side, however, the $\ell_2$ estimation error achieved by these schemes scales as $O(\frac{d}{n2^b})$ under a $b$-bit communication constraint on each sample, where $d$ is the alphabet size of the unknown discrete distribution $p$. This suggests that without additional assumptions on $p$ the error scales *linearly* in $d$, i.e. the introduction of communication constraints introduces a penalty $d$ on the estimation accuracy. This is true even if we allow for interaction between clients [8, 9].

A recent work [10] has moved a step forward from the "global minimax regime" by restricting the target distribution $p$ to be $s$-sparse and showing that the $\ell_2$ error can be reduced to $O(\frac{s \log d}{n2^b})$ in this case, i.e. the error depends on the sparsity $s$ rather than the ambient dimension $d$. However, their scheme heavily relies on the $s$-sparse assumption and requires knowing $s$ beforehand. Therefore, when $s$ is unknown and we need to work with a (potentially loose) upper bound on $s$, the estimation error increases accordingly. In addition, little is known when the target distribution deviates slightly from being exactly $s$-sparse.

In this paper, we argue that all these results can be overly pessimistic, as worst-case notions of complexity and schemes designed to optimize these worst-case notions can be too conservative. Instead, we seek a measure of local complexity that captures the hardness of estimating a specific instance $p$. Ideally, we want a scheme that adapts to the hardness of the problem instead of being tuned to the worst-case scenario; that is, a scheme achieving smaller error when $p$ is "simpler."

**Our contributions**    Motivated by these observations, in this work we consider the *local* minimax complexity of distribution estimation and quantify the hardness of estimating a specific $p$ under communication constraints. In particular, under the $\ell_2$ loss, we show that the local complexity of estimating $p$ is captured by its half-norm[1] $\|p\|_{\frac{1}{2}}$: we propose a two-round interactive scheme that uniformly achieves the $O\left(\max\left(\frac{\|p\|_{1/2}}{n2^b}, \frac{1}{n}\right)\right)$ error under $\ell_2$ loss[2] which requires no prior information on $p$. On the impossibility side, we also show that for any (arbitrarily) interactive scheme, the local minimax error (which is formally defined in Theorem 2.4) must be at least $\Omega\left(\max\left(\frac{\|p\|_{1+\delta/2}}{n2^b}, \frac{1}{n}\right)\right)$ for any $\delta > 0$ when $n$ is sufficiently large.

These upper and the lower bounds together indicate that $\|p\|_{1/2}$ plays a fundamental role in distributed estimation and that the $\lceil \log(\|p\|_{1/2}) \rceil$ bits of communication is both sufficient and necessary to achieve the optimal (centralized) performance when $n$ is large enough. Indeed, this quantity is exactly the Rényi entropy of order 2, i.e. $\mathsf{H}_{1/2}(p) \triangleq \log(\|p\|_{1/2})$, showing that under the $\ell_2$ loss, the correct measure of the local communication complexity at $p$ is the Rényi entropy of $p$.

Compared to the global minimax results where the error scales as $O(\frac{d}{n2^b})$, we see that when we move toward the local regime, the linear dependency on $d$ in the convergence rate is replaced by $\|p\|_{1/2}$. This dimension independent convergence is also empirically verified by our experiments (see Section 3 for more details). Note that $\|p\|_{1/2} < d$, so our proposed scheme is also globally minimax optimal. Moreover, since $\|p\|_{1/2} < \|p\|_0$, our scheme achieves the $O(\frac{s}{n2^b})$ convergence rate under the $s$-sparse model [10], which improves the $O(\frac{s \log(d/s)}{n2^b})$ upper bound in [10] by further shaving off the additional $\log(d/s)$ term (though admittedly, their scheme is designed under the more stringent non-interactive setting). As another immediate corollary, our pointwise upper bounds indicate that 1 bit suffices to attain the performance of the centralized model when the target distribution is highly skewed, such as the (truncated) Geometric distributions and Zipf distributions with degree greater than two.

**Our techniques**    Our proposed two-round interactive scheme is based on a local refinement approach, where in the first round, a standard (global) minimax optimal estimation scheme is applied to localize $p$. In the second round, we use additional $\Theta(n)$ samples (clients), together with the information obtained from the previous round, to locally refine the estimate. The localization-refinement procedure enables us to tune the encoders (in the second round) to the target distribution $p$ and hence attain the optimal pointwise convergence rate uniformly.

On the other hand, our lower bound is based on the quantized Fisher information framework introduced in [11]. However, in order to obtain a local bound around $p$, we develop a new approach that

---

[1]Here we generalize the notion of $q$-norm $\|p\|_q \triangleq \left(\sum_{i=1}^d p_i^q\right)^{1/q}$ to $q \in [0, 1]$.

[2]Our scheme also guarantees pointwise upper bounds on $\ell_1$ or general $\ell_q$ errors. See Theorem 2.2.

first finds the best parametric sub-model containing $p$, and then upper bounds its Fisher information in a neighborhood around $p$. To the best of our knowledge, this is the first impossibility result that allows to capture the local complexity of high dimensional estimation under information constraints and can be of independent interest, e.g. to derive pointwise lower bounds for other estimation models under general information constraints.

## 1.1 Notation and Setup

The general distributed statistical task we consider in this paper can be formulated as follows. Each one of the $n$ clients has local data $X_i \sim p$, where $p \in \mathcal{P}_d$ and $\mathcal{P}_d \triangleq \left\{ p \in \mathbb{R}_+^d \,\middle|\, \sum_j p_j = 1 \right\}$ is the collection of all $d$-dimensional discrete distributions. The $i$-th client then sends a message $Y_i \in \mathcal{Y}$ to the server, who upon receiving $Y^n$ aims to estimate the unknown distribution $p$.

At client $i$, the message $Y_i$ is generated via a *sequentially interactive* protocol; that is, samples are communicated sequentially by broadcasting the communication to all nodes in the system including the server. Therefore, the encoding function $W_i$ of the $i$-th client can depend on all previous messages $Y_1, ..., Y_{i-1}$. Formally, it can be written as a randomized mapping (possibly using shared randomness across participating clients and the server) of the form $W_i(\cdot | X_i, Y^{i-1})$, and the $b$-bit communication constraint restricts $|\mathcal{Y}| \leq 2^b$. As a special case, when $W_i$ depends only on $X_i$ and is independent of the other messages $Y^{i-1}$ for all $i$ (i.e. $W_i(\cdot | X_i, Y^{i-1}) = W_i(\cdot | X_i)$), we say the corresponding protocol is *non-interactive*. Finally, we call the tuple $(W^n, \hat{p}(Y^n))$ an estimation scheme, where $\hat{p}(Y^n)$ is an estimator of $p$. We use $\Pi_{\mathsf{seq}}$ and $\Pi_{\mathsf{ind}}$ to denote the collections of all sequentially interactive and non-interactive schemes respectively.

Our goal here is to design a scheme $(W^n, \hat{p}(Y^n))$ to minimize the $\ell_2$ (or $\ell_1$) estimation error: $r(\ell_2, p, (W^n, \hat{p})) \triangleq \mathbb{E}[\|p - \hat{p}(Y^n)\|_2^2]$, for all $p \in \mathcal{P}_d$, as well as characterizing the best error achievable by any scheme in $\Pi_{\mathsf{seq}}$. We note that while our impossibility bounds hold for any scheme in $\Pi_{\mathsf{seq}}$, the particular scheme we propose uses only one round of interaction. For $\ell_1$ error, we replace $\|\cdot\|_2^2$ in the above expectations with $\|\cdot\|_1$. In this work, we mainly focus on the regime $1 \ll d \ll n$, aiming to characterize the statistical convergence rates when $n$ is sufficiently large.

## 1.2 Related works

Estimating discrete distributions is a fundamental task in statistical inference and has a rich literature [12–15]. Under communication constraints, the optimal convergence rate for discrete distribution estimation was established in [4–8, 16] for the non-interactive setting, and [8, 9] for the general (blackboard) interactive model. The recent work [10] considers the same task under the $s$ sparse assumption for the distribution under communication or privacy constraints. However, all these works study the global minimax error and focus on minimizing the worst-case estimation error. Hence the resultant schemes are tuned to minimize the error in the worst-case which may be too pessimistic for most real-world applications.

A slightly different but closely related problem is distributed estimation of distributions [16–20] and heavy hitter detection under local differential privacy (LDP) constraints [21–24]. Although originally designed for preserving user privacy, some of these schemes can be made communication efficient. For instance, the scheme in [20] suggests that one can use 1 bit communication to achieve $O(d/n)$ $\ell_2$ error (which is global minimax optimal); the non-interactive tree-based schemes proposed in [22, 23] can be cast into a 1-bit frequency oracle, but since the scheme is optimized with respect to $\ell_\infty$ error, directly applying their frequency oracle leads to a sub-optimal convergence $O\left(\frac{s(\log d + \log n)}{n}\right)$ in $\ell_2$.

In the line of distributed estimation under LDP constraint, the recent work [25] studies *local* minimax lower bounds and shows that the local modulus of continuity with respect to the variation distance governs the rate of convergence under LDP. However, the notion of the local minimax error in [25] is based on the two-point method, so their characterized lower bounds may not be uniformly attainable in general high-dimensional settings.

**Organization** The rest of the paper is organized as follows. In Section 2, we present our main results, including pointwise upper bounds and an (almost) matching local minimax lower bound. We provide examples and experiments in Section 3 to demonstrate how our pointwise bounds improve

upon previous global minimax results. In Section 4, we introduce our two-round interactive scheme that achieves the pointwise bound. The analysis of this scheme, as well as the proof of the upper bounds, is given in Section 5. Finally, in Section 6 we provide the proof of the local minimax lower bound.

## 2 Main results

Our first contribution is the design of a two-round interactive scheme (see Section 4 for details) for the problem described in the earlier section. The analysis of the convergence rate of this scheme leads to the following pointwise upper bound on the $\ell_2$ error:

**Theorem 2.1 (Local $\ell_2$ upper bound)** *For any $b \leq \lfloor \log_2 d \rfloor$, there exist a sequentially interactive scheme $(W^n, \check{p}) \in \Pi_{\mathsf{seq}}$, such that for all $p \in \mathcal{P}_d$,*

$$r\left(\ell_2, p, (W^n, \check{p})\right) \leq C_1 \frac{1}{n} + C_2 \frac{\|p\|_{\frac{1}{2}}}{n2^b} + C_3 \frac{d^3 \log(nd)}{(n2^b)^2} \asymp \frac{\|p\|_{\frac{1}{2}} + o_n(1)}{n2^b}, \tag{1}$$

*where $C_1, C_2, C_3 > 0$ are some universal constants (which are explicitly specified in Section 5). This implies that as long as $n = \Omega\left(\frac{d^3 \log d}{2^b \|p\|_{1/2}}\right)$, $r\left(\ell_2, p, (W^n, \check{p})\right) = O\left(\max\left(\frac{\|p\|_{1/2}}{n2^b}, \frac{1}{n}\right)\right)$.*

In addition to the $\ell_2$ error, by slightly tweaking the parameters in our proposed scheme, we can obtain the following pointwise upper bound on the $\ell_1$ (i.e. the total variation) error:

**Theorem 2.2 (Local $\ell_1$ upper bound)** *For any $b \leq \lfloor \log_2 d \rfloor$, there exists a sequentially interactive scheme $\left(\tilde{W}^n, \check{p}\right) \in \Pi_{\mathsf{seq}}$, such that for all $p \in \mathcal{P}_d$,*

$$r\left(\ell_1, p, \left(\tilde{W}^n, \check{p}\right)\right) \leq C_1 \sqrt{\frac{\|p\|_{\frac{1}{2}}}{n}} + C_2 \sqrt{\frac{\|p\|_{\frac{1}{3}}}{n2^b}} + C_3 \sqrt{\frac{d^4 \log(nd)}{(n2^b)^2}} \asymp \sqrt{\frac{\|p\|_{\frac{1}{3}} + o_n(1)}{n2^b}},$$

*where $C_1, C_2, C_3 > 0$ are some universal constants. Hence as long as $n = \Omega\left(\frac{d^4 \log d}{2^b \|p\|_{1/3}}\right)$, $r\left(\ell_1, p, (W^n, \check{p})\right) = O\left(\max\left(\sqrt{\frac{\|p\|_{1/3}}{n2^b}}, \sqrt{\frac{\|p\|_{1/2}}{n}}\right)\right)$.*

Theorem 2.1 implies that the convergence rate of the $\ell_2$ error is dictated by the half-norm of the target distribution $p$ while Theorem 2.2 implies that the $\ell_1$ error is dictated by the one-third norm of the distribution. In general, we can optimize the encoding function with respect to any $\ell_q$ loss for $q \in [1, 2]$ and the quantity that determines the convergence rate becomes the $\frac{q}{q+2}$-norm (see Remark C.1 for details). We also remark that the large second-order terms in Theorem 2.1 and Theorem 2.2 are generally inevitable. See Appendix A.1 for a discussion. The proofs of Theorem 2.1 and Theorem 2.2 are given in Section 5 and Appendix C respectively.

Next, we complement our achievability results with the following minimax lower bounds.

**Theorem 2.3 (Global minimax lower bound)** *For any (possibly interactive) scheme, it holds that*

$$\inf_{(W^n, \hat{p}) \in \Pi_{\mathsf{seq}}} \sup_{p : \|p\|_{1/2} \leq h} \mathbb{E}_{X \sim p}\left[\|\hat{p}(W^n(X^n)) - p\|_2^2\right] = \Omega\left(\frac{h}{n2^b}\right).$$

Note that the lower bound in Theorem 2.3 can be uniformly attained by our scheme even if no information on the target set of distributions, i.e. the parameter $h$ is available, indicating that our proposed scheme is global minimax optimal. The proof can be found in Section D.

**Theorem 2.4 (Local minimax lower bound)** *Let $p \in \mathcal{P}'_d \triangleq \left\{ p \in \mathcal{P}_d \big| \frac{1}{2} < p_1 < \frac{2}{3} \right\}$. Then for any $\delta > 0, B \geq \sqrt{\frac{\|p\|_{1/2}}{d 2^b}}$, as long as $n = \Omega\left( \frac{d^3 \log d}{\|p\|_{1/2}} \right)$, it holds that[3]*

$$\inf_{(W^n, \hat{p}) \in \Pi_{\text{seq}}} \sup_{p' : \|p' - p\|_\infty \leq \frac{B}{\sqrt{n}}} \mathbb{E}_{p'}\left[ \|\hat{p}\left( W^n(X^n) \right) - p'\|_2^2 \right] \geq \frac{1}{n 2^b} \max\left( c_\delta \|p\|_{\frac{1+\delta}{2}} , \frac{c_1 \|p\|_{\frac{1}{2}}}{\log d} \right) + \frac{c_2}{n},$$

*for some $c_\delta, c_1, c_2 > 0$.*

**Remark 2.1** *Note that in Theorem 2.4 the $\ell_\infty$ neighborhood $\left\{ p' : \|p' - p\|_\infty \leq \frac{B}{\sqrt{n}} \right\}$ is strictly smaller than the following $\ell_2$ neighborhood: $\left\{ p' : \|p' - p\|_2 \leq \sqrt{\|p\|_{1/2}/(2^b n)} \right\}$ (since $\mathcal{B}_{d, \ell_\infty}(\frac{1}{\sqrt{d}}) \subset \mathcal{B}_{d, \ell_2}(1)$). Therefore, in Theorem 2.4 we present a slightly stronger statement.*

Theorem 2.4 indicates that our proposed scheme is (almost) optimal when $n$ is sufficiently large and that $\|p\|_{1/2}$ is the fundamental limit for the estimation error. This conclusion implies that under $\ell_2$ loss, the right measure of the hardness of an instance $p$ is its half-norm, or equivalently its Rényi entropy of order $1/2$. The proof of Theorem 2.4 is given in Section 6.

**Corollary 2.1** *When $n$ is sufficiently large, $\lceil \log(\|p\|_{1/2}) \rceil$ bits of communication is both sufficient and necessary to achieve the convergence rate of the centralized setting under $\ell_2$ loss. Similarly, $\lceil \log(\|p\|_{1/3}) \rceil$ bits are sufficient for $\ell_1$ loss.*

In addition, observe that the quantities $\log(\|p\|_{1/2})$ and $\frac{1}{2}\log\left( (p\|_{1/3}) \right)$ are the Rényi entropies of order $1/2$ and $1/3$, denoted as $\mathsf{H}_{1/2}(p)$ and $\mathsf{H}_{1/3}(p)$, respectively. In other words, the communication required to achieve the optimal (i.e. the centralized) rate is determined by the Rényi entropy of the underlying distribution $p$.

Finally, we remark that when the goal is to achieve the centralized rate with minimal communication (instead of achieving the best convergence rate for a fixed communication budget as we have assumed so far), the performance promised in the above corollary can be achieved without knowing $\mathsf{H}_{1/2}(p)$ beforehand. See Appendix A.2 for a discussion.

# 3 Examples and experiments

Next, we demonstrate by several examples and experiments that our results can recover earlier global minimax results and can significantly improve them when the target distribution is highly skewed.

**Corollary 3.1 (s-sparse distributions)** *Let $P_{s,d} \triangleq \left\{ p \in [0,1]^d \big| \sum p_i = 1, \|p\|_0 \leq s \right\}$. Then as long as $n$ large enough, there exists interactive schemes $(W^n, \check{p})$ and $(\tilde{W}^n, \check{p})$ such that $r\left( \ell_2, p, (W^n, \check{p}) \right) = O(\frac{s}{n 2^b})$ and $r\left( \ell_1, p, \left( \tilde{W}^n, \check{p} \right) \right) = O(\sqrt{\frac{s}{n 2^b}})$.*

The above result shows that our scheme is minimax optimal over $\mathcal{P}_{d,s}$. This recovers and improves (by a factor of $\log d/s$) the results from [10] on the $\ell_1$ and $\ell_2$ convergence rates for $s$-sparse distributions[4].

**Corollary 3.2 (Truncated geometric distributions)** *Let $p \triangleq \mathsf{Geo}_{\beta,d}$ be the truncated geometric distribution. That is, for all $\beta \in (0,1]$ and $k \in [d]$, $\mathsf{Geo}_{\beta,d}(k) \triangleq \frac{1-\beta}{\beta(1-\beta^d)} \beta^k$. Then if $n = \Omega\left( d^3 \log d / 2^b \right)$,*

$$r\left( \ell_2, p, (W^n, \hat{p}) \right) = O\left( \frac{1}{n 2^b} \frac{(1 + \sqrt{\beta})(1 - \sqrt{\beta}^d)}{(1 - \sqrt{\beta})(1 + \sqrt{\beta}^d)} \right) = O\left( \max\left( \frac{1}{n(1 - \sqrt{\beta}) 2^b}, \frac{1}{n} \right) \right).$$

---

[3]Indeed, the lower bound holds for blackboard interactive schemes [26], a more general class of interactive schemes than $\Pi_{\text{seq}}$. See [8] for a discussion of blackboard schemes.

[4]We remark, however, that their scheme is *non-interactive* and has smaller minimum sample size requirement.

This result shows that if $\beta$ is constant for the truncated geometric distribution, 1 bit suffices to achieve the centralized convergence rate $O\left(\frac{1}{n}\right)$ in this case. Note that this is a significant improvement over previous minimax bounds on the the $\ell_2$ error which are $O(\frac{d}{n2^b})$ [4]. This suggests that the corresponding minimax optimal scheme is suboptimal by a factor of $d$ when the target distribution is truncated geometric. Our results suggest that the $\ell_2$ error should not depend on $d$ at all, and Figure 1 provides empirical evidence to justify this observation.

**Corollary 3.3 (Truncated Zipf distributions with $\lambda > 2$)** *Let $p \triangleq \mathsf{Zipf}_{\lambda,d}$ be a truncated Zipf distribution with $\lambda \geq 2$. That is, for $k \in [d]$, $\mathsf{Zipf}_{\lambda,d}(k) \triangleq \frac{k^{-\lambda}}{\sum_{k'=1}^{d}(k')^{-\lambda}}$. Then the local complexity of $p$ is characterized by $\|p\|_{1/2} = \Theta\left(\left(\frac{1-d^{-\lambda/2+1}}{1-d^{-\lambda+1}}\right)^2\right) = \Theta(1)$. So in this case, $r\left(\ell_2, p, (W^n, \hat{p})\right) = O\left(\frac{1}{n}\right)$, as long as $n = \Omega\left(d^3 \log d/2^b\right)$.*

We leave the complete characterization for all $\lambda > 0$ to Section E in appendix. Note that since both $\mathsf{Geo}_{\beta,d}$ and $\mathsf{Zipf}_{\lambda,d}$ are not sparse distributions, [10] cannot be applied here. Therefore, the best previously known scheme for these two cases is the global minimax optimal scheme achieving an $O\left(\frac{d}{n2^b}\right)$ $\ell_2$ error. Again, our results suggest that this is suboptimal by a factor of $d$.

In Figure 1, we empirically compare our scheme with [4] (which is globally minimax optimal). In the left figure, we see that the error of our scheme is an order of magnitude smaller than the minimax scheme and remains almost the same under different values of $d$. We illustrate that more clearly in the right figure, where we fix $n$ and increase $d$. It can be observed that the error of our scheme remains bounded when $d$ increases, while the error of the minimax scheme scales linearly in $d$. This phenomenon is justified by Corollary 3.2.

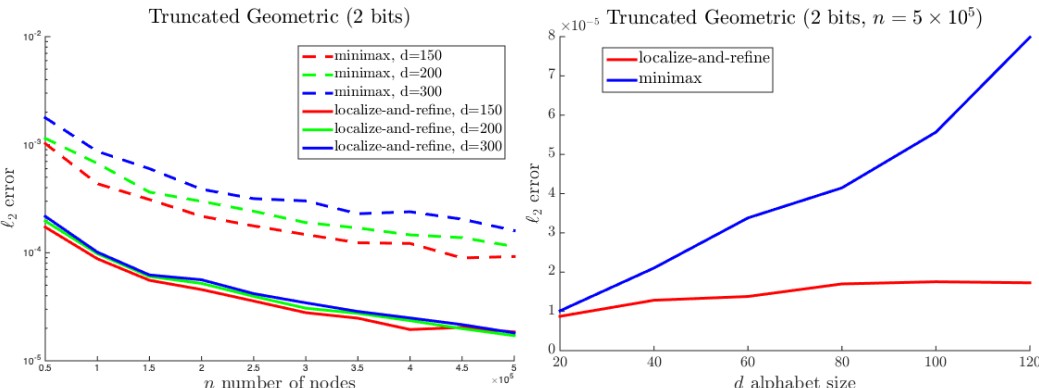

Figure 1: Comparisons between our scheme (labeled as 'localize-and-refine') and the globally minimax optimal scheme (labeled as 'minimax') [4]. The underlying distribution is set to be truncated geometric distribution with $\beta = 0.8$.

## 4 The localization-refinement scheme

In this section, we introduce our two-round localization-refinement scheme and show how it locally refines a global minimax estimator $\hat{p}$ to obtain a pointwise upper bound.

The first round of our two-round estimation scheme (the localization phase) is built upon the global minimax optimal scheme [4], in which symbols are partitioned into $\lfloor d/(2^b - 1)\rfloor$ subsets each of size $(2^b - 1)$, and each of the first $n/2$ clients is assigned one of these subsets (each subset is assigned to $(n/2)/(d/(2^b - 1))$ clients). The client reports its observation only if it is in its assigned subset by using $b$ bits. This first round allows us to obtain a coarse estimate $\hat{p}$ with error $\mathbb{E}[\|p - \hat{p}\|_2^2] \asymp \sum_j p_j/n2^b = O(d/n2^b)$. Then in the second round (the refinement phase), we locally refine the estimate by adaptively assigning more "resources" (i.e. samples), according to $\hat{p}$ obtained from the first round, to the more difficult symbols (i.e. $j \in [d]$ with larger error

| **Algorithm 1:** uniform grouping [4] (at client $i$) | **Algorithm 2:** localize-and-refine (at client $i$) |
|---|---|
| **Input:** $X_i \in [d], b \in \mathbb{N}$ | **Input:** $X_i \in [d], b \in \mathbb{N}, \hat{p}$ |
| Compute $M = d/(2^b - 1)$; | $(\mathcal{G}_1, ..., \mathcal{G}_d) \leftarrow \texttt{gen\_group}(\hat{p}, d), Y_i \leftarrow 0$ ; |
| $m \leftarrow i \bmod M, Y_i \leftarrow 0$; | Compute $\mathcal{J}_i \triangleq \{j \in [d] \| i \in \mathcal{G}_j\}$ ; |
| **if** $X_i \in \left[(m-1)(2^b-1)+1 : m(2^b-1)\right]$ **then** | **if** $X_i \in \mathcal{J}_i$ **then** |
| $\quad\mid\quad Y_i \leftarrow X_i \bmod (2^b - 1)$ | $\quad\mid\quad Y_i \leftarrow$ the ranking of $X_i$ in $\mathcal{J}_i$ |
| **end** | **end** |
| **return** $Y_i$ | **return** $Y_i$ |

$\mathbb{E}\left[(\hat{p}_j - p_j)^2\right]$). In particular, under the $\ell_2$ loss, the number of samples assigned to symbol $j$ will be roughly in proportional to $\sqrt{p_j}$. It turns out that the local refinement step can effectively mitigate the estimation error, enabling us to move from a worst-case bound to a pointwise bound.

**Round 1 (localization):** In the first round, the first $n/2$ clients collaboratively produce a coarse estimate $\hat{p} = (\hat{p}_1, ..., \hat{p}_j)$ for the target distribution $p$ via the grouping scheme described above (this is the minimax optimal scheme proposed in [4], see Algorithm 1 for details). Note that in general, this scheme can be replaced by any non-interactive minimax optimal scheme.

**Round 2 (local refinement):** Upon obtaining $\hat{p}$ from the first round, the server then computes $\pi(\hat{p}) : \mathcal{P}_d \to \mathcal{P}_d$, where the exact definition of $\pi$ depends on the loss function. $\pi$ will be used in the design of the rest $n/2$ clients' encoders. In particular, for $\ell_2$ loss, we set $\pi_j(p) \triangleq \frac{\sqrt{p_j}}{\sum_{k \in [d]} \sqrt{p_k}}$; for $\ell_1$ loss, we set $\pi_j(p) \triangleq \frac{\sqrt[3]{p_j}}{\sum_{k \in [d]} \sqrt[3]{p_k}}$. For notational convenience, we denote $\pi(\hat{p})$ as $\hat{\pi} = (\hat{\pi}_1, ..., \hat{\pi}_d)$.

To design the encoding functions of the remaining $n/2$ clients, we group them into $d$ (possibly overlapping) sets $\mathcal{G}_1, ..., \mathcal{G}_d$ with

$$|\mathcal{G}_j| = n_j \triangleq \frac{n}{2}\left(\min\left(1, (2^b - 1)\left(\frac{\hat{\pi}_j}{4} + \frac{1}{4d}\right)\right)\right), \tag{2}$$

for all $j \in [d]$.

We require the grouping $\{\mathcal{G}_j\}$ to satisfy the following properties: 1) each $\{\mathcal{G}_j\}$ consists of $n_j$ distinct clients, and 2) each client $i$ is contained in at most $(2^b - 1)$ groups. Notice that these requirements can always be attained (see, for instance, Algorithm 3 in Section B). We further write $\mathcal{J}_i$ to be the set of indices of the groups that client $i$ belongs to (i.e. $\mathcal{J}_i \triangleq \{j \in [d] \| i \in \mathcal{G}_j\}$), so the second property implies $|\mathcal{J}_i| \leq 2^b - 1$ for all $i$.

As in the first round, client $i$ will report its observation if it belongs to the subset $\mathcal{J}_i$. Since $|\mathcal{J}_i| \leq 2^b - 1$, client $i$'s report can be encoded in $b$ bits. Note that with this grouping strategy each symbol $j$ is reported by clients in $\mathcal{G}_j$, hence the server can estimate $p_j$ by computing $\check{p}_j(Y^n) \triangleq \frac{\sum_{i \in \mathcal{G}_j} \mathbb{1}_{\{X_i = j\}}}{n_j}$. Note that the size of $\mathcal{G}_j$ is dictated by the estimate for $p_j$ obtained in the first round. See Algorithm 2 for the details of the second round.

**Remark 4.1** *In the above two-round scheme, we see that the local refinement step is crucial for moving from the worst-case performance guarantee to a pointwise one. Therefore we conjecture that the local lower bound in Theorem 2.4 cannot be achieved by any non-interactive scheme.*

## 5 Analysis of the $\ell_2$ error (proof of Theorem 2.1)

In this section, we analyze the estimation errors of the above two-round localization-refinement scheme and prove Theorem 2.1. Before entering the main proof, we give the following lemma that controls the estimation error of between $\hat{p}$ and $\pi$.

**Lemma 5.1** *Let $(W^n, \hat{p})$ be the grouping scheme given in Algorithm 1. Then with probability at least $1 - \frac{1}{nd}$, it holds that $\left|\sqrt{p_j} - \sqrt{\hat{p}_j}\right| \leq \sqrt{\varepsilon_n}$ and $\left|\sqrt[3]{p_j} - \sqrt[3]{\hat{p}_j}\right| \leq \sqrt[3]{\varepsilon_n}$, where $\varepsilon_n \triangleq \frac{3d \log(nd)}{n2^b}$.*

Now consider the scheme described in Section 4. After the first round, we obtain a coarse estimate $\hat{p} = (\hat{p}_1, ..., \hat{p}_j)$. Set $\mathcal{E} \triangleq \bigcap_{j \in [d]} \left\{ \sqrt{\hat{p}_j} \in \left[ \sqrt{p_j} - \sqrt{\varepsilon_n}, \sqrt{p_j} + \sqrt{\varepsilon_n} \right] \right\}$. Then by Lemma 5.1 and taking the union bound over $j \in [d]$, we have $\mathbb{P}\{\mathcal{E}\} \geq 1 - \frac{1}{n}$. In order to distinguish the estimate obtained from the first round to the final estimator, we use $\check{p}_j$ to denote the final estimator.

Now observe that the $\ell_2$ estimation error can be decomposed into

$$\mathbb{E}\left[\|p - \check{p}\|_2^2\right] = \mathbb{P}\{\mathcal{E}^c\}\,\mathbb{E}\left[\|p - \check{p}\|_2^2\Big|\mathcal{E}^c\right] + \mathbb{P}\{\mathcal{E}\}\,\mathbb{E}\left[\|p - \check{p}\|_2^2\Big|\mathcal{E}\right] \overset{(a)}{\leq} \frac{2}{n} + \mathbb{E}\left[\|p - \check{p}\|_2^2\Big|\mathcal{E}\right]$$

where (a) holds since $\|p - \check{p}\|_2^2 \leq 2$ almost surely. Hence it remains to bound $\mathbb{E}\left[\|p - \check{p}\|_2^2\Big|\mathcal{E}\right]$. Next, as described in Section 4, we partition the second $n/2$ clients into $d$ overlapping groups $\mathcal{G}_1, ..., \mathcal{G}_d$ according to (2). The reason we choose $n_j$ in this way is to ensure that 1) for symbols with larger $p_j$ (which implies larger estimation error), we allocate them more samples; 2) every symbol is assigned with at least $\Theta(n/d)$ samples.

Clients in $\mathcal{G}_j$ then collaboratively estimate $p_j$. In particular, client $i$ reports her observation if $X_i \in \mathcal{J}_i \triangleq \{j | i \in \mathcal{G}_j\}$, and the server computes $\check{p}_j(Y^n|\hat{p}) \triangleq \frac{\sum_{i \in \mathcal{G}_j} \mathbb{1}\{X_i = j\}}{n_j} \sim \frac{1}{n_j}\mathrm{Binom}(n_j, p_j)$. Finally, the following lemma controls $\mathbb{E}\left[\|p - \check{p}\|_2^2\Big|\mathcal{E}\right]$, completing the proof of Theorem 2.1.

**Lemma 5.2** *Let $\check{p}$ be defined as above. Then* $\mathbb{E}\left[\|p - \check{p}\|_2^2\Big|\mathcal{E}\right] \leq \frac{6}{n2^b}\left(\sum_{j \in [d]} \sqrt{p_j}\right)^2 + \frac{10d^2\varepsilon_n}{n2^b} + \frac{1}{n}$.

For the $\ell_1$ error, we set $\pi_j(p) \triangleq \frac{\sqrt[3]{p_j}}{\sum_k \sqrt[3]{p_k}}$. Following the similar but slightly more sophisticated analysis, we obtain Theorem 2.2. The detailed proof is left to Section C in appendix.

# 6 The local minimax lower bound (proof of Theorem 2.4)

Our proof is based on the framework introduced in [8], where a global upper bound on the quantized Fisher information is given and used to derive the minimax lower bound on the $\ell_2$ error. We extend their results to the local regime and develop a *local* upper bound on the quantized Fisher information around a neighborhood of $p$.

To obtain a local upper bound, we construct an $h$-dimensional parametric sub-model $\Theta_p^h$ that contains $p$ and is a subset of $\mathcal{P}_d$, where $h \in [d]$ is a tuning parameter and will be determined later. By considering the sub-model $\Theta_p^h$, we can control its Fisher information around $p$ with a function of $h$ and $p$. Optimizing over $h \in [d]$ yield an upper bound that depends on $\|p\|_{\frac{1}{2}}$. Finally, the local upper bound on the quantized Fisher information can then be transformed to the local minimax lower bound on the $\ell_2$ error via van Tree's inequality [27].

Before entering the main proof, we first introduce some notation that will be used through this section. Let $(p_{(1)}, p_{(2)}, ..., p_{(d)})$ be the sorted sequence of $p = (p_1, p_2, ..., p_d)$ in the non-increasing order; that is, $p_{(i)} \geq p_{(j)}$ for all $i > j$. Denote $\pi : [d] \to [d]$ as the corresponding sorting function[5], i.e. $p_{(i)} = p_{\pi(i)}$ for all $i \in [d]$.

**Constructing the sub-model $\Theta_p^h$** We construct $\Theta_p^h$ by "freezing" the $(d - h)$ smallest coordinates of $p$ and only letting the largest $(h - 1)$ coordinates to be free parameters. Mathematically, let

$$\Theta_p^h \triangleq \left\{ (\theta_2, \theta_3, ..., \theta_h) \big| \pi^{-1}\left(\theta_1, \theta_2, ..., \theta_h, p_{(h+1)}, p_{(h+2)}, ..., p_{(d)}\right) \in \mathcal{P}_d \right\}, \tag{3}$$

where $\theta_1 = 1 - \sum_{i=2}^h \theta_i - \sum_{i=h+1}^d p_i$ is fixed when $(\theta_2, ..., \theta_h)$ are determined. For instance, if $p = \left(\frac{1}{16}, \frac{1}{8}, \frac{1}{2}, \frac{1}{16}, \frac{1}{4}\right)$ (so $d = 5$) and $h = 3$, then the corresponding sub-model is

$$\Theta_p^h \triangleq \left\{ (\theta_2, \theta_3) \Big| \left(\frac{1}{16}, \theta_3, \theta_1, \frac{1}{16}, \theta_2\right) \in \mathcal{P}_d \right\}.$$

---

[5] With a slight abuse of notation, we overload $\pi$ so that $\pi\left((p_1, ..., p_d)\right) \triangleq (p_{\pi(1)}, ..., p_{\pi(d)})$

**Bounding the quantized Fisher information**  Next, under this model, we control the quantized Fisher information in the following lemma.

**Lemma 6.1**  *Let $W$ be any $b$-bit quantization scheme and $I_W(\theta)$ be the Fisher information of $Y$ at $\theta$ where $Y \sim W(\cdot|X)$ and $X \sim p_\theta$. Let $0 < B \leq \frac{p_{(h)}}{3}$ and $p \in \mathcal{P}'_d \triangleq \left\{ p \in \mathcal{P}_d \middle| \frac{1}{2} < p_1 < \frac{5}{6} \right\}$. Define the neighborhood $\mathcal{N}_{B,h}(p) \triangleq \theta(p) + [-B, B]^h$ (note that under this definition, $\mathcal{N}_{B,h}(p)$ must be contained in $\Theta_p^h$). Then*

$$\forall \theta' \in \mathcal{N}_{B,h}(p), \ \mathsf{Tr}\left(I_W(\theta')\right) \leq 2^b \left( 6h + \frac{3}{2p_{(h)}} \right).$$

**Bounding the $\ell_2$ error**  Applying [8, Theorem 3] on $\mathcal{N}_{B,h}(p)$, we obtain

$$\sup_{\theta' \in \mathcal{N}_{B,h}(p)} \mathbb{E}\left[ \left\| \hat{\theta} - \theta' \right\|_2^2 \right] \geq \frac{h^2}{n2^b \left( 6h + \frac{3}{2p_{(h)}} \right) + \frac{h\pi^2}{B^2}} \geq \frac{h^2 p_{(h)}}{10n2^b + \frac{10hp_{(h)}}{B^2}} \geq \frac{h^2 p_{(h)}}{20n2^b}, \quad (4)$$

where the second inequality is due to $hp_{(h)} \leq 1$, and the third inequality holds if we pick $B \geq \sqrt{\frac{hp_{(h)}}{n2^b}}$. Notice that in order to satisfy the condition $\mathcal{N}_{B,h}(p) \subseteq \Theta_p^h$, $B$ must be at most $\frac{p_{(h)}}{3}$, so we have an implicit sample size requirement: $n$ must be at least $\frac{3h}{2^b p_{(h)}}$.

**Optimizing over $h$**  Finally, we maximize $h^2 p_{(h)}$ over $h \in [d]$ to obtain the best lower bound. The following simple but crucial lemma relates $h^2 p_{(h)}$ to $\|p\|_{\frac{1}{2}}$.

**Lemma 6.2**  *For any $p \in \mathcal{P}_d$ and $\delta > 0$, it holds that*

$$\|p\|_{\frac{1}{2}} \geq \max_{h \in [d]} h^2 p_{(h)} \geq \max \left( C_\delta \|p\|_{\frac{1+\delta}{2}}, C\|p\|_{\frac{1}{2}} / \log d \right),$$

*for $C_\delta \triangleq \left( \frac{\delta}{1+\delta} \right)^{\frac{2}{1+\delta}}$ and a universal constant $C$ small enough.*

Picking $h^* = \operatorname{argmax}_{h \in [d]} h^2 p_{(h)}$ and by Lemma 6.2 and (4), we obtain that for all $\hat{\theta}$

$$\sup_{\theta' \in \mathcal{N}_{B_n,h^*}(p)} \mathbb{E}\left[ \left\| \hat{\theta} - \theta' \right\|_2^2 \right] \geq \max \left( C'_\delta \frac{\|p\|_{\frac{1+\delta}{2}}}{n2^b}, C' \frac{\|p\|_{\frac{1}{2}}}{n2^b \log d} \right),$$

as long as $p \in \mathcal{P}'_d$ and $B_n = \sqrt{\frac{h^* p_{(h^*)}}{n2^b}} \overset{(a)}{\leq} \sqrt{\frac{d}{n2^b \|p\|_{\frac{1}{2}}}}$, where (a) holds due to the second result of Lemma 6.2 and $h^* \leq d$. In addition, the sample size constraint that $n$ must be larger than $\frac{3h^*}{2^b p_{(h^*)}}$ can be satisfied if $n = \Omega \left( \frac{d^3 \log d}{2^b \|p\|_{\frac{1}{2}}} \right)$ since $\frac{h^*}{p_{(h^*)}} \leq \frac{(h^*)^3 \log d}{C\|p\|_{\frac{1}{2}}} \leq \frac{d^3 \log d}{C\|p\|_{\frac{1}{2}}}$, where the first inequality is due to Lemma 6.2 and the second one is due to $h^* \leq d$. The proof is complete by observing that

$$\inf_{(W^n, \hat{p})} \sup_{p' : \|p' - p\|_\infty \leq B_n} \mathbb{E}\left[ \|\hat{p} - p'\|_2^2 \right] \geq \inf_{(W^n, \hat{\theta})} \sup_{\theta' \in \mathcal{N}_{B_n,h^*}(p)} \mathbb{E}\left[ \left\| \hat{\theta} - \theta' \right\|_2^2 \right].$$

## 7  Conclusion and open problems

We have investigated distribution estimation under $b$-bit communication constraints and characterized the local complexity of a target distribution $p$. We show that under $\ell_2$ loss, the half-norm of $p$ dictates the convergence rate of the estimation error. In addition, to achieve the optimal (centralized) convergence rate, $\Theta(\mathsf{H}_{1/2}(p))$ bits of communication is both necessary and sufficient.

Many interesting questions remain to be addressed, including investigating if the same lower bound can be achieved by non-interactive schemes, deriving the local complexity under general information constraints (such as $\varepsilon$-local differential privacy constraint), and extending this results to the distribution-free setting (i.e. the frequency estimation problem).

## Acknowledgments

This work was supported in part by a Google Faculty Research Award, a National Semiconductor Corporation Stanford Graduate Fellowship, and the National Science Foundation under grants CCF-1704624 and NeTS-1817205.

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
