# A  Additional Remarks

## A.1  Second-order terms in the upper and lower bounds

In terms of the convergence rate, Theorem 2.1 and Theorem 2.2 admit large second-order terms which may dominate the MSEs when $n \preceq d^3$. However, our results improve the sample complexity for the high-accuracy regime. More precisely, as a direct corollary of Theorem 2.1, the $\ell_2$ sample complexity is $O\left(\max\left(\frac{1}{\varepsilon}, \frac{\|p\|_{1/2}}{n2^b}, \frac{d^3 \log d}{\sqrt{\varepsilon} \log(1/\varepsilon)4^b}\right)\right)$. In addition, the $n \geq \mathsf{poly}(d)$ requirement is necessary for all of the previous global minimax schemes. For instance, all the minimax upper and lower bounds for distribution estimation (without any additional assumptions on the target distribution) in previous works [4,5,7,8] require $n \geq d^2$.

That being said, with additional prior knowledge on the target distribution such as sparse/near-sparse assumptions, we can easily improve our two-stage scheme by replacing the uniform grouping localization step with other sparse estimation schemes [28], and the resulting sample size requirement can be decreased to $n \geq \mathsf{poly}(s, \log d)$.

## A.2  Achieving centralized convergence without the knowledge of $\|p\|_{1/2}$

We note that when the goal is to achieve the centralized rate with minimal communication (instead of achieving the best convergence rate for a fixed communication budget as we have assumed so far), the performance promised in the above corollary can be achieved without knowing $\mathsf{H}_{1/2}(p)$ beforehand. We can do that by modifying our proposed scheme to include an initial round for estimating $\mathsf{H}_{1/2}(p)$. More precisely, we can have the first $n/2$ clients communicate only 1 bit about their sample and estimate $\mathsf{H}_{1/2}(p)$ within 1 bit accuracy. When $n$ is sufficiently large (e.g. $n = \tilde{\Omega}(d^3)$), this estimation task will be successful with negligible error probability. Then the remaining $n/2$ of clients can implement the two-round scheme we propose to estimate $p$ (i.e. the scheme described in Section 4) by using $\hat{\mathsf{H}}_{1/2}(p) + 1$ bits per client. Under this strategy, we are guaranteed to achieve the centralized performance while each client communicates no more than $\mathsf{H}_{1/2}(p) + 2$ bits.

# B  Adaptive grouping algorithm

In this section, we describes the details of the adaptive grouping algorithm used in Section 4 and Algorithm 2.

---

**Algorithm 3:** `gen_groups`

---

**Input:** $\hat{p}$, d
**Output:** $\mathcal{G}_1, ..., \mathcal{G}_d$
Compute $\hat{\pi}_j$ and $n_j$ according to (2);
$\{n_{\sigma(1)}, ..., n_{\sigma(d)}\} \leftarrow \mathsf{sort}(\{n_1, ..., n_d\})$ ;      // Sort $\{n_j\}$ in non-increasing order.
$\mathcal{G}_1, ..., \mathcal{G}_d \leftarrow \emptyset$;
**for** $i \in [n+1 : 2n]$ **do**
    **for** $t \leq 2^b$ **do**
        **for** $j \leq d$ **do**
            **if** $\left|\mathcal{G}_{\sigma(j)}\right| < n_{\sigma(j)}$ **then**
                $\mathcal{G}_{\sigma(j)} \leftarrow i$;
                $t \leftarrow t + 1$;
            **else**
                $j \leftarrow j + 1$;
            **end**
        **end**
    **end**
    $t \leftarrow 2^b$;
**end**
**return** $\mathcal{G}_1, ..., \mathcal{G}_d$

---

# C  Proof of Theorem 2.2

Again, consider the scheme in Section 4 but with $\pi_j(p) \triangleq \frac{\sqrt[3]{p_j}}{\sum_{k \in [d]} \sqrt[3]{p_k}}$. Let $\mathcal{J}_\alpha \triangleq \{j \in [d] | p_j \geq \alpha\}$ and

$$\mathcal{E}_1 \triangleq \bigcap_{j \in [d]} \left\{ \sqrt[3]{\hat{p}_j} \in \left[ \sqrt[3]{p_j} - \sqrt[3]{\varepsilon_n}, \sqrt[3]{p_j} + \sqrt[3]{\varepsilon_n} \right] \right\},$$

where $\alpha > 0$ will be determined later. Notice that by Lemma 5.1, $\mathbb{P}\{\mathcal{E}_1\} > 1 - \frac{1}{n}$, so we have the following bounds on the $\ell_1$ error:

$$\mathbb{E}\left[\|p - \check{p}\|_{\text{TV}}\right] = \mathbb{P}\{\mathcal{E}_1^c\} \mathbb{E}\left[\sum_{j \in [d]} |p_j - \check{p}_j| \Big| \mathcal{E}_1^c\right] + \mathbb{P}\{\mathcal{E}_1\} \mathbb{E}\left[\sum_{j \in [d]} |p_j - \check{p}_j| \Big| \mathcal{E}_1\right]$$

$$\leq \frac{2}{n} + \mathbb{E}\left[\sum_{j \in [d]} |p_j - \check{p}_j| \Big| \mathcal{E}_1\right].$$

As in the $\ell_2$ case, the server computes $\check{p}_j \triangleq \frac{1}{n_j}\text{Binom}(n_j, p_j)$. For the ease of analysis, we partition $[d]$ into three disjoint subsets:

$$\mathcal{J}^+ \triangleq \{j \in [d] | n_j = n\}, \ \mathcal{J}_\alpha \triangleq \{j \in [d] \setminus \mathcal{J}^+ | p_j \geq \alpha\} \text{ and } \mathcal{J}^c \triangleq [d] \setminus (\mathcal{J}^+ \cup \mathcal{J}_\alpha),$$

where $\alpha > 0$ will be specified later. For each subset, we will apply different lower bound on $n_j$:

- for $j \in \mathcal{J}^+$, we have $n_j = \frac{n}{2}$;

- for $j \in \mathcal{J}_\alpha$, we use $n_j \geq \frac{n2^b \hat{\pi}_j}{4}$;

- for $j \in \mathcal{J}^c$, we use $n_j \geq \frac{n2^b}{4d}$.

We can then compute the estimation error by

$$\mathbb{E}\left[\sum_{j \in [d]} |p_j - \check{p}_j| \Big| \mathcal{E}_1\right] \leq \sum_{j \in [d]} \sqrt{\mathbb{E}\left[(p_j - \check{p}_j)^2 \big| \mathcal{E}_1\right]} \overset{(a)}{\leq} \sum_{j \in [d]} \sqrt{\frac{p_j}{n_j}}$$

$$\overset{(b)}{\leq} \sum_{j \in \mathcal{J}^+} \sqrt{\frac{2p_j}{n}} + \sum_{j \in \mathcal{J}_\alpha} \sqrt{\frac{4p_j}{n2^b \hat{\pi}_j}} + \sum_{j \in \mathcal{J}^c} \sqrt{\frac{4dp_j}{n2^b}}$$

$$\overset{(c)}{\leq} \sqrt{\frac{2\|p\|_{\frac{1}{2}}}{n}} + \sum_{j \in \mathcal{J}_\alpha} \sqrt{\frac{4p_j}{n2^b \hat{\pi}_j}} + \sqrt{\frac{4d^3\alpha}{n2^b}}$$

$$\overset{(d)}{=} \sqrt{\frac{2\|p\|_{\frac{1}{2}}}{n}} + \sqrt{\frac{4}{n2^b}} \left(\sqrt{\sum_{j' \in [d]} \sqrt[3]{\hat{p}_{j'}}}\right) \left(\sum_{j \in \mathcal{J}_\alpha} \sqrt{\frac{p_j}{\sqrt[3]{\hat{p}_j}}}\right) + \sqrt{\frac{4d^3\alpha}{n2^b}}$$

$$\overset{(d)}{\leq} \sqrt{\frac{2\|p\|_{\frac{1}{2}}}{n}} + \sqrt{\frac{4}{n2^b}} \left(\sqrt{\sum_{j' \in [d]} \left(\sqrt[3]{p_{j'}} + \sqrt[3]{\varepsilon_n}\right)}\right) \left(\sum_{j \in \mathcal{J}_\alpha} \sqrt{\frac{p_j}{\sqrt[3]{p_j} - \sqrt[3]{\varepsilon_n}}}\right) + \sqrt{\frac{4d^3\alpha}{n2^b}}, \quad (5)$$

where (a) holds since conditioned on $\mathcal{E}_1$, $\check{p} \sim \frac{1}{n}\text{Binom}(n_j, p_j)$, (b) holds by the definition of $n_j$, (c) is due to Cauchy-Schwartz inequality, (d) follows from the definition of $\hat{\pi}_j$ and finally (d) is due to

Lemma 5.1. If we pick $\alpha = 8\varepsilon_n$, then we can further bound (5) by

$$\sqrt{\frac{2\|p\|_{\frac{1}{2}}}{n}} + \sqrt{\frac{4}{n2^b}} \left(\sqrt{\sum_{j'\in[d]}\left(\sqrt[3]{p_j} + \sqrt[3]{\varepsilon_n}\right)}\right)\left(\sum_{j\in\mathcal{J}_\alpha}\sqrt{\frac{p_j}{\sqrt[3]{p_j}-\sqrt[3]{\varepsilon_n}}}\right) + \sqrt{\frac{4d^3\alpha}{n2^b}}$$

$$\overset{(a)}{\leq} \sqrt{\frac{2\|p\|_{\frac{1}{2}}}{n}} + \sqrt{\frac{16}{n2^b}\left(\sum_{j\in[d]}\sqrt[3]{p_j}\right)^3} + \sqrt{\frac{16d\sqrt[3]{\varepsilon_n}}{n2^b}\left(\sum_{j\in[d]}\sqrt[3]{p_j}\right)^2} + \sqrt{\frac{32d^3\varepsilon_n}{n2^b}}$$

$$\overset{(b)}{\leq} \sqrt{\frac{2\|p\|_{\frac{1}{2}}}{n}} + \sqrt{\frac{16}{n2^b}\left(\sum_{j\in[d]}\sqrt[3]{p_j}\right)^3} + \sqrt{\frac{48d^3\varepsilon_n}{n2^b}}$$

$$\leq C_1\sqrt{\frac{\|p\|_{\frac{1}{2}}}{n}} + C_2\sqrt{\frac{\|p\|_{\frac{1}{3}}}{n2^b}} + C_3\sqrt{\frac{d^3\varepsilon_n}{n2^b}},$$

where (a) holds since $\sqrt{a+b} \leq \sqrt{a} + \sqrt{b}$ and in (b) we use the following Young's inequality: $a^{\frac{1}{3}}b^{\frac{2}{3}} \leq \frac{a}{3} + \frac{2b}{3} \leq a + b$.

**Remark C.1** *For general $\ell_q$ error with $q \in [1,2]$, we pick $\pi_j(p) = p_j^{1-\frac{2}{q+1}}(\sum_k p_j^{1-\frac{2}{q+1}})^{-1}$, and the local $\ell_q$ error under our scheme becomes*

$$\mathbb{E}\left[\sum_{j\in[d]}(p_j - \check{p}_j)^q\right] \preceq \frac{\left(\sum_{j\in[d]}p_j^{q/(q+2)}\right)^{(q+2)/2} + o_n(1)}{(n2^b)^{q/2}}.$$

## D  Proof of Theorem 2.3

Let $s \triangleq \lfloor h \rfloor$ and observe that $\mathcal{P}_{s,d} \triangleq \{p \in \mathcal{P}_d | \|p\|_0 \leq s\} \subseteq \left\{p \in \mathcal{P}_d \Big| \|p\|_{1/2} \leq h\right\}$. Then the proof is complete since

$$\inf_{(W^n,\hat{p})\in\Pi_{\text{seq}}} \sup_{p:\|p\|_{1/2}\leq h} \mathbb{E}_{X\sim p}\left[\|\hat{p}(W^n(X^n)) - p\|_2^2\right]$$

$$\geq \inf_{(W^n,\hat{p})\in\Pi_{\text{seq}}} \sup_{p:\mathcal{P}_{s,d}} \mathbb{E}_{X\sim p}\left[\|\hat{p}(W^n(X^n)) - p\|_2^2\right]$$

$$\succeq \frac{s}{n2^b},$$

where the last inequality holds by [10].

## E  A complete characterization for the local complexity of Zipf distributions

Below we characterize the local complexity of Zipf distributions for all regime $\lambda > 0$. Interestingly, as $\lambda$ decreases, we see different dependency on $d$.

**Corollary E.1 (Truncated Zipf distributions with $\lambda \geq 2$)** *Let $p \triangleq \text{Zipf}_{\lambda,d}$ be a truncated Zipf distribution with $\lambda \geq 2$. That is, for $d, k \in \mathbb{N}$, $\text{Zipf}_{\lambda,d}(k) \triangleq \frac{\frac{1}{k^\lambda}\mathbb{1}_{\{k\leq d\}}}{\sum_{k'=1}^d \frac{1}{(k')^\lambda}}$. Then the skewness of $p$ is characterized by*

*1. $\|p\|_{1/2} = \Theta\left(\left(\frac{1-d^{-\lambda/2+1}}{1-d^{-\lambda+1}}\right)^2\right) = \Theta(1)$, if $\lambda > 2$. In this case, $r(\ell_2, p, (W^n, \hat{p})) = O\left(\frac{1}{n}\right)$.*

*2. $\|p\|_{1/2} = \Theta\left(\left(\frac{\log d}{1-d^{-\lambda+1}}\right)^2\right) = \Theta\left(\log^2 d\right)$, if $\lambda = 2$, so $r(\ell_2, p, (W^n, \hat{p})) = O\left(\frac{\log^2 d}{n2^b} \vee \frac{1}{n}\right)$.*

3. $\|p\|_{1/2} = \Theta\left(\left(\frac{d^{-\lambda/2+1}}{1-d^{-\lambda+1}}\right)^2\right) = \Theta\left(d^{2-\lambda}\right)$, if $1 < \lambda < 2$. So $r\left(\ell_2, p, (W^n, \hat{p})\right) = O\left(\frac{d^{2-\lambda}}{n2^b} \vee \frac{1}{n}\right)$.

4. $\|p\|_{1/2} = \Theta\left(\left(\frac{d^{-\lambda/2+1}}{\log d}\right)^2\right) = \Theta\left(\frac{d}{\log^2 d}\right)$, if $\lambda = 1$. So $r\left(\ell_2, p, (W^n, \hat{p})\right) = O\left(\frac{d}{n2^b \log^2 d} \vee \frac{1}{n}\right)$.

5. $\|p\|_{1/2} = \Theta\left(\left(\frac{d^{-\lambda/2+1}}{d^{-\lambda+1}}\right)^2\right) = \Theta\left(d\right)$, if $\lambda < 1$. So $r\left(\ell_2, p, (W^n, \hat{p})\right) = O\left(\frac{d}{n2^b} \vee \frac{1}{n}\right)$.

# F   Proof of technical lemmas

## F.1   Proof of Lemma 5.1

Let $(W^n, \hat{p})$ be the naive grouping scheme introduced by [4]. Then $\hat{p}_j \sim \frac{1}{n'}\mathsf{Binom}(n', p_j)$, where $n' \triangleq \frac{n}{2d}$. By the Chernoff bound, with probability at least $1 - \frac{1}{nd}$,

$$\hat{p}_j \in \left[p_j - \sqrt{\frac{3d\log(nd)p_j}{n2^b}}, p_j + \sqrt{\frac{3d\log(nd)p_j}{n2^b}}\right] = \left[p_j - \sqrt{\varepsilon_n p_j}, \, p_j + \sqrt{\varepsilon_n p_j}\right]. \quad (6)$$

The first result follows from

$$\left|\sqrt{p_j} - \sqrt{\hat{p}_j}\right| = \left|\frac{p_j - \hat{p}_j}{\sqrt{p_j} + \sqrt{\hat{p}_j^+}}\right| \leq \frac{\sqrt{\varepsilon_n p_j}}{\sqrt{p_j}} = \sqrt{\varepsilon_n}$$

and taking union bound over $j \in [d]$. To prove the second result, for each $j \in [d]$, we consider two cases.

- If $p_j \geq \varepsilon_n$, then it holds with probability at least $1 - 1/nd$ that

$$\left|\sqrt[3]{\hat{p}_j} - \sqrt[3]{p_j}\right| = \frac{|\hat{p}_j - p_j|}{\hat{p}_j^{2/3} + (\hat{p}_j p_j)^{1/3} + p_j^{2/3}} \overset{(a)}{\leq} \frac{\sqrt{\varepsilon_n p_j}}{p_j^{2/3}} = \sqrt{\varepsilon_n} p_j^{-1/6} \overset{(b)}{\leq} \varepsilon_n^{1/3},$$

  where (a) holds with probability at least $1 - 1/nd$ due to (6), and (b) holds since by assumption $p_j \geq \varepsilon_n$.

- If $p_j \leq \varepsilon_n$, then since $\hat{p}_j \geq 0$ almost surely, it suffices to control

$$\begin{aligned}
\mathbb{P}\left\{\sqrt[3]{\hat{p}_j} > \sqrt[3]{p_j} + \sqrt[3]{\varepsilon_n}\right\} &\leq \mathbb{P}\left\{\hat{p}_j > p_j + \varepsilon_n\right\} \\
&\leq \mathbb{P}\left\{|\hat{p}_j - p_j| > \varepsilon_n\right\} \\
&\overset{(a)}{\leq} \mathbb{P}\left\{|\hat{p}_j - p_j| > \sqrt{\varepsilon_n p_j}\right\} \\
&\overset{(b)}{\leq} \frac{1}{nd},
\end{aligned}$$

  where (a) holds since $p_j \leq \varepsilon_n$, and (b) holds by (6).

The proof is complete since for both cases we have $\left|\sqrt[3]{\hat{p}_j} - \sqrt[3]{p_j}\right| \leq \sqrt[3]{\varepsilon_n}$ with probability at least $1 - \frac{1}{nd}$.

## F.2   Proof of Lemma 5.2

To analyze the error, we partition $[d]$ into three disjoint subsets:

$$\mathcal{J}^+ \triangleq \left\{j \in [d] | n_j = n\right\}, \, \mathcal{J}_\alpha \triangleq \left\{j \in [d] \setminus \mathcal{J}^+ \big| p_j \geq \alpha\right\} \text{ and } \mathcal{J}^c \triangleq [d] \setminus \left(\mathcal{J}^+ \cup \mathcal{J}_\alpha\right).$$

for some $\alpha > 0$ that will be specified later. For each subset, we use different lower bound on $n_j$:

- for $j \in \mathcal{J}^+$, we have $n_j = \frac{n}{2}$;
- for $j \in \mathcal{J}_\alpha$, we use $n_j \geq \frac{n2^b \hat{\pi}_j}{4}$;

- for $j \in \mathcal{J}^c$, we use $n_j \geq \frac{n2^b}{4d}$.

We can then compute the estimation error by

$$
\mathbb{E}\left[\sum_{j \in [d]} (p_j - \check{p}_j)^2 \Big| \mathcal{E}\right] \overset{(a)}{=} \sum_{j \in [d]} \frac{p_j(1 - p_j)}{n_j}
$$

$$
\overset{(b)}{\leq} \sum_{j \in \mathcal{J}^+} \frac{2p_j}{n} + \sum_{j \in \mathcal{J}_\alpha} \frac{4p_j}{n2^b \hat{\pi}_j} + \sum_{j \in \mathcal{J}^c} \frac{4dp_j}{n2^b}
$$

$$
\overset{(c)}{\leq} \frac{2}{n} + \sum_{j \in \mathcal{J}_\alpha} \frac{4p_j}{n2^b \hat{\pi}_j} + \frac{4d^2\alpha}{n2^b}
$$

$$
\overset{(d)}{=} \frac{2}{n} + \frac{4}{n2^b}\left(\sum_{j' \in [d]} \sqrt{\hat{p}_{j'}}\right)\left(\sum_{j \in \mathcal{J}_\alpha} \frac{p_j}{\sqrt{\hat{p}_j}}\right) + \frac{4d^2\alpha}{n2^b}
$$

$$
\overset{(e)}{\leq} \frac{2}{n} + \frac{4}{n2^b}\left(\sum_{j' \in [d]} \sqrt{p_{j'}} + \sqrt{\varepsilon_n}\right)\left(\sum_{j \in \mathcal{J}_\alpha} \frac{p_j}{\sqrt{p_j} - \sqrt{\varepsilon_n}}\right) + \frac{4d^2\alpha}{n2^b},
$$

where (a) holds since $\check{p}_j$ follows binomial distribution, (b) holds by the definition of $n_j$, (c) is due to the definition of $\mathcal{J}_\alpha$, (d) is due to the definition of $\hat{\pi}_j$, and finally (e) is from Lemma 5.1.

In particular, if we pick $\alpha = 4\varepsilon_n$, then for all $j \in \mathcal{J}_\alpha$, $\sqrt{\varepsilon_n} = \frac{\sqrt{\alpha}}{2} \leq \frac{\sqrt{p_j}}{2}$. Then (e) above can be further controlled by

$$
\frac{4}{n2^b}\left(\sum_{j' \in [d]} \sqrt{p_{j'}} + \sqrt{\varepsilon_n}\right)\left(\sum_{j \in \mathcal{J}_\alpha} \frac{p_j}{\sqrt{p_j} - \sqrt{\varepsilon_n}}\right) + \frac{4d^2\alpha}{n2^b} + \frac{2}{n}
$$

$$
\leq \frac{8}{n2^b}\left(\sum_{j \in [d]} \sqrt{p_j}\right)\left(\sum_{j \in \mathcal{J}_\alpha} \sqrt{p_j}\right) + \frac{8d\sqrt{\varepsilon_n}}{n2^b}\left(\sum_{j \in \mathcal{J}_\alpha} \sqrt{p_j}\right) + \frac{16d^2\varepsilon_n}{n2^b} + \frac{2}{n}
$$

$$
\overset{(a)}{\leq} \frac{12}{n2^b}\left(\sum_{j \in [d]} \sqrt{p_j}\right)^2 + \frac{20d^2\varepsilon_n}{n2^b} + \frac{2}{n}
$$

$$
\leq \frac{C_1}{n} + C_2 \frac{\|p\|_{\frac{1}{2}}}{n2^b} + C_3 \frac{d^2\varepsilon_n}{n2^b},
$$

where in (a) we use the fact that $a^2 + b^2 \geq 2ab$.

### F.3 Proof of Lemma 6.1

Recall that under the model $\Theta_p^h$, the score function

$$
S_\theta(x) \triangleq (S_{\theta_2}(x), ..., S_{\theta_h}(x)) \triangleq \left(\frac{\partial \log p(x|\theta)}{\partial \theta_2}, ..., \frac{\partial \log p(x|\theta)}{\partial \theta_h}\right)
$$

can be computed as

$$
S_{\theta_i}(x) = \begin{cases} \frac{1}{\theta_i}, & \text{if } x = \pi(i), \ 2 \leq i \leq h \\ -\frac{1}{\theta_1}, & \text{if } x = \pi(1) \\ 0, & \text{otherwise} \end{cases}
$$

The next lemma shows that to bound the quantized Fisher information, it suffices to control the variance of the score function.

**Lemma F.1 (Theorem 1 in [8])** *Let $W$ be any $b$-bit quantization scheme and $I_W(\theta)$ is the Fisher information of $Y$ at $\theta$ where $Y \sim W(\cdot|X)$ and $X \sim p_\theta$. Then for any $\theta \in \Theta \subseteq \mathbb{R}^h$,*

$$
\mathrm{Tr}\left(I_W(\theta)\right) \leq \min\left(I_X(\theta), 2^b \max_{\|u\|_2 \leq 1} \mathrm{Var}\left(\langle u, S_\theta(X)\rangle\right)\right).
$$

Therefore, for any unit vector $u = (u_2, ..., u_h)$ with $\|u\|_2 = 1$, we control the variance as follows:

$$\text{Var}\left(\langle u, S_\theta(X)\rangle\right) \overset{(a)}{=} \sum_{i=1}^{h} \theta_i \left(\sum_{j=2}^{h} u_j S_{\theta_j}(\pi(i))\right)^2$$

$$= \theta_1 \left(\sum_{j=2}^{h} u_j\right)^2 \left(\frac{1}{\theta_1}\right)^2 + \sum_{i=2}^{h} \theta_i u_i^2 \frac{1}{\theta_i^2}$$

$$= \frac{\left(\sum_{j=2}^{h} u_j\right)^2}{\theta_1} + \sum_{j=2}^{h} \frac{u_i^2}{\theta_i}$$

$$\leq \frac{h}{\theta_1} + \frac{1}{\min_{j \in \{2,...,h\}} \theta_j}, \tag{7}$$

where (a) holds since the score function has zero mean. This allows us to upper bound $I_W(\theta)$ in a neighborhood around $\theta(p)$, where $\theta(p)$ is the location of $p$ in the sub-model $\Theta_p^h$, i.e.

$$\theta(p) = (\theta_2(p), ..., \theta_h(p)) \triangleq (p_{(2)}, ..., p_{(h)}).$$

In particular, for any $0 < B \leq \frac{p_{(h)}}{3}$ and $p \in \mathcal{P}_d' \triangleq \left\{p \in \mathcal{P}_d \big| \frac{1}{2} < p_1 < \frac{5}{6}\right\}$, the neighborhood $\mathcal{N}_{B,h}(p) \triangleq \theta(p) + [-B, B]^h$ must be contained in $\Theta_p^h$. In addition, for any $\theta' \in \mathcal{N}_{B,h}(p)$, it holds that

$$\theta_1' \geq \theta_1(p) - \frac{h p_{(h)}}{3} \geq \frac{1}{6},$$

where the second inequality holds since 1) $\theta_1(p) = p_{(1)} \geq \frac{1}{2}$ by our definition of $\mathcal{P}_d'$, and 2) $\frac{h p_{(h)}}{3} \leq \frac{\sum_{i=1}^h p_{(1)}}{3} \leq \frac{1}{3}$. We also have

$$\min_{j \in \{2,...,h\}} \theta_j' \geq \min_{j \in \{2,...,h\}} \theta_j(p) - \frac{p_{(h)}}{3} \geq \frac{2p_{(h)}}{3}.$$

Therefore (7) implies for any $\theta' \in \mathcal{N}_{B,h}(p)$,

$$\text{Var}\left(\langle u, S_\theta'(X)\rangle\right) \leq 6h + \frac{3}{2p_{(h)}}.$$

Together with Lemma F.1, we arrive at

$$\forall \theta' \in \mathcal{N}_{B,h}(p), \ \text{Tr}\left(I_W(\theta')\right) \leq 2^b \left(6h + \frac{3}{2p_{(h)}}\right).$$

### F.4 Proof of Lemma 6.2

We prove by contradiction.

1. For the first inequality, assume by contradiction that $\|p\|_{\frac{1}{2}} \geq \max_{h \in [d]} h^2 p_{(h)}$ does not hold. Then there must exists some $\tilde{h} \in [d]$ such that

$$\tilde{h}^2 p_{\tilde{h}} > \|p\|_{\frac{1}{2}} \iff \sqrt{p_{\tilde{h}}} > \frac{\sum_{j \in [d]} \sqrt{p_j}}{\tilde{h}}. \tag{8}$$

However, this implies

$$\|p\|_{\frac{1}{2}} \overset{(a)}{=} \left(\sum_{j=1}^{d} \sqrt{p_j}\right)^2 \geq \left(\sum_{j=1}^{\tilde{h}} \sqrt{p_{(j)}}\right)^2 \overset{(b)}{\geq} \left(\sum_{j=1}^{\tilde{h}} \sqrt{p_{(\tilde{h})}}\right)^2 \overset{(c)}{>} \left(\sum_{j=1}^{d} \sqrt{p_j}\right)^2,$$

where (a) is by definition, (b) holds since $p_{(\tilde{h})} \leq p_{(j)}$ for all $j \leq \tilde{h}$, and (c) is due to (8). This yields contradiction.

2. Secondly, assume by contradiction that for all $h$, $h^2 p_{(h)} < \left(\frac{\delta}{1+\delta}\right)^{\frac{2}{1+\delta}} \left(\sum_{j\in[d]} p_j^{\frac{1+\delta}{2}}\right)^{\frac{2}{1+\delta}}$.
   This would imply

   $$p_{(h)}^{\frac{1+\delta}{2}} < \left(\frac{\delta}{1+\delta}\right) \frac{\sum_{j\in[d]} p_j^{\frac{1+\delta}{2}}}{h^{1+\delta}}$$

   $$\iff \sum_h p_{(h)}^{\frac{1+\delta}{2}} = \sum_{j\in[d]} p_j^{\frac{1+\delta}{2}} < \left(\frac{\delta}{1+\delta}\right) \left(\sum_{j\in[d]} p_j^{\frac{1+\delta}{2}}\right) \sum_h \frac{1}{h^{1+\delta}}$$

   $$\iff \sum_h \frac{1}{h^{1+\delta}} > 1 + \frac{1}{\delta},$$

   which cannot occur since $\sum_{h\in\mathbb{N}} \frac{1}{h^{1+\delta}} \le 1 + \frac{1}{\delta}$.

3. Finally assume $\max_{h\in[d]} h^2 p_{(h)} \ge C \frac{\|p\|_{\frac{1}{2}}}{\log d}$ does not hold. Then it would imply that for all $h$

   $$\sqrt{p_{(h)}} < C \frac{\|p\|_{\frac{1}{2}}}{\log d} \frac{1}{h} \iff \sum_{h=1}^d \sqrt{p_{(h)}} = \|p\|_{\frac{1}{2}} < C \frac{\|p\|_{\frac{1}{2}}}{\log d} \left(\sum_{h=1}^d \frac{1}{h}\right)$$

   $$\iff \left(\sum_{h=1}^d \frac{1}{h}\right) > \frac{1}{C} \log d.$$

   Picking $C$ small enough yields contradiction.