# OpenReview forum: "Pointwise Bounds for Distribution Estimation under Communication Constraints"
_NeurIPS.cc/2021/Conference — NeurIPS 2021 Poster_

### Official Review · Reviewer_qr7y · 2021-07-02

**Rating:** 6
**Confidence:** 3

**Summary:**

This paper considers the estimation of a discrete distribution P on an alphabet of size d under communication constraints. Each local client holds a single observation from P and transmits no more than b bits to the central server. A two-round interactive scheme is proposed and achieves nearly optimal estimation error. The estimation error depends on the Rényi entropy of P instead of the global parameter d. The minimax lower bound is also derived in this paper.

**Limitations And Societal Impact:**

The limitations are discussed in the paper. The potential negative societal impacts are not available as the paper is mostly on the theoretical aspects.

**Main Review:**

Overall the problem considered is of broad interest and the results in this paper are solid. The fundamental limit on the communication resources in terms of the Rényi entropy is one significant contribution of the current paper. The main concern is on the additional terms of Theorems 2.1 and 2.2 (hidden from the abstract when n is sufficiently large). The quantitative implication is n=poly(d), which appears unrealistic and should be properly justified. With so many clients, the total communication costs are still high. This issue also concerns Theorem 2.4 where the local region scales with n. More issues in Theorem 2.4: the set $\mathcal{P}_d'$ is introduced without context; the term $\delta>0$ is not really arbitrary as $c_\delta$ vanishes with $\delta$, so it should be optimized if possible.

**Time Spent Reviewing:**

5

---

> ### Author Response · Authors · 2021-08-10
> **Response to Reviewer qr7y**
>
> We thank the reviewer for the valuable comments and suggestions.
>
> > “The main concern is on the additional terms of Theorems 2.1 and 2.2.”
>
> On the practical side, there are many applications that fit into the parameter regime that $n = \mathsf{poly}(d)$, see the first point in the response to Reviewer 2 (JieB) for a detailed discussion. We will justify the assumptions that $n = \mathsf{poly}(d)$ in the revision. In addition, the $ n = \mathsf{poly}(d)$ requirement appears in the literature when there are no  additional assumptions on the target distribution. For instance, all the minimax upper and lower bounds in previous works [Han et. al 2018a, 2018b; Barnes 2019] require $n \geq d^2$.
>
> On the other hand, we believe that the $n = \tilde{\Omega}(d^3)$ threshold is fundamental as the same threshold also naturally appears in the lower bound analysis and seems to be unavoidable. Therefore, we conjecture that the error rate may behave differently in the small sample regime $n \preceq d^3$, and our upper and lower bounds characterize the optimal rate for the large sample regime $n \succeq d^3$. Bridging the gap for the small sample regimes is left as future work.
>
>
> > “The local region in Theorem 2.4 scales with n.”
>
> We remark that for the local minimax theory, the region should shrink with $n$; otherwise, the lower bound becomes a global minimax result and cannot capture the instance-specific hardness in the estimation task. Obtaining the local minimax results is usually more difficult than characterizing the global minimax lower bound, and this is one of our main contributions. Also note that the region shrinks at the rate $O(1/\sqrt{n})$, which is the same as the classical Hajek-Lecam’s local asymptotic minimax theory.
>
> > “The set $\mathcal{P}’_d$ is introduced without context.”
>
> Choosing the particular form of $\mathcal{P}’_d$ is to facilitate the analysis of the lower bounds. This assumption can be circumvented with a more sophisticated analysis, but we decided to introduce it in order to make the proof easier to follow. We will add a discussion on how this assumption comes into play.
>
> > “Since $c_\delta$ vanishes with $\delta$, it should be optimized if possible.”
>
> We remark that although $c_\delta$ vanishes with $\delta$, $c_\delta>0$ for any $\delta>0$ and is independent of $p$ or $d$, and hence our lower bounds can be interpreted as follows: for any fixed but arbitrarily small $\delta>0$, the local minimax MSE scales at least at the rate of $O\left( \frac{\lVert p \rVert_{\frac{1+\delta}{2}}}{n}\right)$. Note that $\lVert p \rVert_{\frac{1+\delta}{2}}$ depends on $p$ and may grow with increasing $d$. Since $\delta$  can be taken arbitrarily small, the scaling of our lower can be made arbitrarly close to the  upper bound.
>
> On the other hand, each $\delta$ provides a different (pointwise) lower bound on the local minimax error; in other words, for a given $p$  $\delta$ can be optimized to give the largest lower bound, however the resultant lower bound may not have a clean closed-form expression. In addition to the $\delta$-dependent lower bounds, we also provide another lower bound that has a $\log(d)$ gap. We believe that the current form of the lower bounds  is the best way to state our impossibility result and illustrate that our upper bound is (almost) order-optimal.
>
> In the subsequent revision, we will make sure $c_\delta$ is explicitly stated in the theorem.
>
> References:
> [Han et. al, 2018a] "Distributed statistical estimation of high dimensional and nonparametric distributions", ISIT 2018
> [Han et. al, 2018b] "Geometric lower bounds for distributed parameter estimation under communication constraints", COLT 2018
> [Barnes et. al, 2019] "Lower bounds for learning distributions under communication constraints via fisher information", JMLR 2020

---

> > ### Comment · Reviewer_qr7y · 2021-08-17
> > **I maintain the score**
> >
> > Thanks, I have read the responses.
> >
> > I still think $n\ge d^3$ is restrictive (also much worse than $d^2$ you mentioned). Even in applications where the number of samples is larger than the domain size, $d^3$ easily blows up. As other quantitative results are available, it is important to include proper discussions such as why you believe the transition happens at $d^3$.
> >
> > For Theorem 2.4, I understand the local region $n^{-1/2}$. The concern is also on the $n\ge d^3$ in the assumption.
> >
> > I don't agree with "the scaling of our lower can be made arbitrarly close to the upper bound" due to the extra vanishing $c_\delta$. I do appreciate the lower bound $\Vert p\Vert_{1/2} /\log d $ though.

---

### Official Review · Reviewer_d6xD · 2021-07-17

**Rating:** 8
**Confidence:** 4

**Summary:**

The paper studies distributed estimation of a discrete, $d$-dimensional law $p$ under communication constraints (wherein each client may only send $b$ bits) in a sequential model (wherein clients are ordered, and are aware of messages sent by earlier clients when preparing their message).

The authors describe what is essentially a two-stage scheme that first obtains a crude estimate of the law using a minimax scheme. This estimate yields a sense of which elements of the law are large, and subsequent communications are designed that provide more samples for such elements, the precise weighting of which depends on the error metric being controlled. As a result, if the number of clients is large enough ($n \gg d^3\log d$ for squared-$\ell_2$ error), the resulting estimate for the squared-$\ell_2$ criterion has error decaying as $\max( \\| p\\|_{1/2}2^{-b} , 1)/n$. Both stages of the scheme use the grouping idea of Han et al [4].

The above scheme is complemented by a lower bound for the $\ell_2$ criterion, which shows that under the same large $n$ condition, a matching lower bound but with a $(1+\delta)/2$ norm instead of the $1/2$ norm, lending a measure of tightness to the achievability analysis. This is derived via a Fisher information technique due to Barnes et al, and a nice construction of a parametric family of discrete laws in which the lowest $d-h$ masses are fixed. Under some auxiliary conditions, the resulting bound depends linearly on $h^2 p_{(h)},$ where $p_{(h)}$ is the $h$th biggest entry of $p$, and this quantity is characterised in terms of the $1/2$ and $(1+\delta)/2$ norms.

Together, these results almost characterise the bit-rate needed by each client to ensure optimal estimation, specifically the $1/2$-Renyi entropy of $p.$ This immediately leads to improved rates for spase, and for highly skewed laws.


**Limitations And Societal Impact:**

The limitations with respect to $n$ should be discussed more deeply, as described above.

**Main Review:**

Post Rebuttal: thanks for the clarifications. I think my view basically remains the same.

----


I think the paper discusses a relevant problem, and the proposed scheme and lower bound are quite elegant. Pointwise results are always particularly interesting, and the bound of this paper are novel, and offer a broad insight into distributed estimation. It's particularly nice that these offer direct improvements for cases such as sparsity.

Perhaps the biggest weakness of the analysis is the large error terms, which require a large number of clients to be effective. Equivalently, the results of the paper do not offer improvements on sample complexity of the tasks for a given bitrate. On the whole this is not a deal breaker - I think there may well be regimes where one does have access to a large number of clients, none of which can communicate much (although whether such a situation would admit the sequential model is a little questionable). I think that the paper should discuss this aspect explicitly [*], and contextualise the analysis by describing application domains where such an assumption is reasonable (and ideally also where it is not). In addition, a discussion of whether the authors believe that these terms are an artifact of the analysis or more fundamental would also be useful to the community.

[*] while the authors are conscientious about stating the bounds completely and acknowleding the requirements on $n$, I think this is not enough.

---

Clarity: The paper is mostly well written, and the approach is well explained. The related work described is, to my knowledge, adequate, and the results are properly contextualised with respect to these (barring a frank discussion of the size of $n$, as I discussed above). I think the paper can benefit from explaining where the particular form of the $n_j$ comes from. This seems not hard to motivate, at least for the $\ell_2$ squared error - using the explanation on line 267, the net error behaves as $\sum p_j/n_j$, and since no client can account for more than $2^b-1$ symbols, there's a sum constraint on the $n_j$, and minimising under it leads to the choice of $n_j \propto \sqrt{p_j}$.

----

A few minor points that need correction:

- Equation (2) is missing a factor of $2$ somewhere (there are only $n/2$ clients).

- The event $\mathcal{E}$ on 257 should be an intersection of each of the $\sqrt{\hat{p_j}}$ being close, not a union. Similarly for $\mathcal{E}_1$ in the appendix.

- Equation (6) is not quite right - the Binomial tails flatten at the extremes, due to which if $n' p$ is small, then concentration at $\sqrt{\varepsilon_n p}$ as claimed is not right, and instead we have something like (Bernstein's inequality) $\hat{p} \in p \pm \max(\epsilon_n, \sqrt{\varepsilon_n p})$. The argument can be fixed by separately considering the case $p < \varepsilon_n \implies \hat{p} < 2\varepsilon_n $ which directly gives $|\sqrt{p} - \sqrt{\hat{p}}| \le 3\sqrt{\varepsilon_n}.$ Also, for completeness' sake, the usual statement of Hoeffding's inequality is not sensitive to the variance, and something like Bernstein's or Bennet's is needed.

- The proof for $C_\delta = \delta^{2/(1+\delta)} $ in Lemma 6.2 does not quite follow (in fact $\sum_{h \in \mathbb{N}} 1/h^{1 + \delta} > 1/\delta  $). This is not a big deal, since the bound $\sum_{h \in \mathbb{N}} 1/h^{1 + \delta} \le 1 + 1/\delta  $ is true, and this in turn gives $C_\delta = (\delta/(1+\delta) )^{2 / (1 + \delta)},$ which has the same qualitative behaviour for the relevant small $\delta$.



**Time Spent Reviewing:**

4h

---

> ### Author Response · Authors · 2021-08-10
> **Response to Reviewer d6xD**
>
> We thank the reviewer for the thorough feedback, useful suggestions, and for appreciating our techniques.
>
> > “The biggest weakness of the analysis is the large error terms, and the results of the paper do not offer improvements on sample complexity.”
>
> We acknowledge that in terms of the convergence rate, our scheme consists of a large second-order term which may dominate MSE when $n \preceq d^3$. However, we would also like to add that our results do improve the sample complexity for the high-accuracy regime. More precisely, as a direct corollary of Theorem 2.2, the sample complexity of our scheme is $O\left( \max\left( \frac{1}{\varepsilon}, \frac{\lVert p \rVert_{1/2}}{2^b \varepsilon}, \frac{d^3\log(d)}{\sqrt{\varepsilon}\log(1/\varepsilon)4^b} \right) \right)$, which characterizes the optimal sample complexity when $\varepsilon$ is small enough (i.e. for the high-accuracy regime). We will add details to discuss this point in the revision.
>
>
> > “There may well be regimes where one does have access to a large number of clients, none of which can communicate much.”
>
> We agree with the reviewer that there are many applications that fit into this scenario, see the first point in the response to Reviewer 2 (JieB) for a detailed discussion. We will justify the assumptions that $n = \mathsf{poly}(d)$ and explicitly discuss the limitations on both theoretical and practical sides in the revision.
>
>
> > “Whether these terms are an artifact of the analysis or more fundamental.”
>
> Thanks for bringing up this excellent point. We believe that the $n = \tilde{\Omega}(d^3)$ threshold is fundamental as the same threshold also naturally appears in the lower bound analysis and seems to be unavoidable. Therefore, we conjecture that the error rate may behave differently in the small sample regime $n \preceq d^3$, and our upper and lower bounds only characterize the optimal rate for the large sample regime $n \succeq d^3$. Bridging this gap is of both theoretical and practical interest and is left as future work. Again, we will make sure this point is properly discussed in the revision.
>
>
> > “The paper can benefit from explaining where the particular form of the $n_j$ comes from.”
>
> We agree with the reviewer and will elaborate more on the intuition/motivation behind the particular choice of $n_j$.
>
>
> > “A few minor points that need correction.”
>
> We thank the reviewer for carefully catching the typos/minor mistakes and providing excellent suggestions to fix them. We will correct them in the revision. In particular, regarding the third point (i.e. the correctness of equation (6)), what we were actually using is the Chernoff bound for Bernoulli random variables instead of Hoeffding’s inequality (sorry for the very misleading typo). Chernoff bound is sensitive to the variance, so the resulting bound has an additional $p_j$ term on the numerator, compared to the resulting bound from Hoeffding’s inequality.

---

### Official Review · Reviewer_JieB · 2021-07-21

**Rating:** 6
**Confidence:** 4

**Summary:**

The paper studies estimating an unknown discrete distribution over $d$ symbols from its samples observed under $b$-bit communication constraint. The main result states that for a sufficiently large sample size $n$, the optimal $\ell_2$ error is essentially
$\Theta \left(
\frac{\lVert p \rVert_{1/2}}{n2^b}
\lor
\frac1n
\right)$.
The paper also obtains an upper bound on the $\ell_1$ learning rate without a matching lower bound.
The optimal interactive scheme first finds a coarse estimate of the underlying distribution with an existing minimax approach,
then refines the result using this estimate and an independent sample.

**Limitations And Societal Impact:**

The authors discussed the limitations of their work and marked N/A for the potential negative societal impact of their work.

**Main Review:**

Pros:
1. The paper is well-written and provides appropriate examples and plots.
2. I appreciate the authors' efforts to develop the proposed learning scheme that achieves near-optimal performance for every distribution. The idea is also natural enough to lead to a simple algorithm.
3. The results also imply a logarithmic improvement over an existing scheme on $s$-sparse distributions, though under a different setting.

Cons:
1. It seems that the results, including Theorems 2.1, 2.2., and 2.4, hold only for large sample sizes $n$, e.g., $n\gtrsim d^3$. I wonder if this is the case for practical applications. As per my understanding, these types of distribution learning applications often involve large alphabets.
2. The minimax lower bound in Theorem 2.4 considers the supremum over distributions $p'$ satisfying $\lVert p' - p\rVert_\infty \le \sqrt{\frac{\lVert p\rVert_{1/2}}{n d 2^{b}}}$. I wonder if the $\ell_{1/2}$-norm values remain close to that of $p$ or not, which is critical to the merits of the lower bound.
3. The optimality of the results on the $\ell_1$ norm is unclear. From my perspective, $\ell_1$ distance is probably a better measure for distribution learning as it directly relates to classification errors in hypothesis selection.
4. In line 191, the paper claims that the scheme is minimax optimal over $\mathcal P_{d, s}$. I can see the upper bound, but not a (nearly matching lower bound for $s$-sparse distributions. Note that the prior work [10] considers the problem under the non-interactive setting, which is more restrictive.
5. Line 193 states that "since $\beta$ is constant," I wonder why this is the case and note that Corollary 3.2 assumes only $\beta\in(0, 1/3)$. Also, Line 204 claims that the results in [10] do not apply to geometric and Zipf distributions. My thought is that these distributions have small "effective support sizes," learnable with a standard minimax algorithm.

Additional suggestion:
1. It might be better if the authors can shorten the abstract as it has 20 lines of text.
2. It seems reasonable to consider experiments in more practical settings, such as those mentioned in the Introduction.





**Time Spent Reviewing:**

5

---

> ### Author Response · Authors · 2021-08-10
> **Response to Reviewer JieB**
>
> We thank the reviewer for the valuable comments and suggestions.
> > “Theorems 2.1, 2.2., and 2.4, hold only for large sample sizes $n$, e.g. $n \succeq d^3$”
>
> On the practical side, there are many applications where the number of samples is larger than the domain size, for example, emoji prediction task, hit song prediction, or sensor networks, etc. In some of these scenarios (such as sensor networks), the communication cost may be extremely stringent as each distributed sensor has limited power to transmit the signal it observed, and hence reducing the communication is crucial and of practical interest.
>
> On the other hand, as discussed in the reply to Reviewer 1 (PqZC), the $ n \geq \mathsf{poly}(d)$ requirement appears in the literature. For instance, all the minimax upper and lower bounds for distribution estimation (without any additional assumptions on the target distribution) in previous works [Han et. al 2018a, 2018b; Barnes 2019] require $n \geq d^2$.
>
> We would also like to remark that with additional prior knowledge on the target distribution such as sparse/near-sparse assumptions, we can easily improve our two-stage scheme by replacing the uniform grouping localization step with other sparse estimation schemes (such as [Acharya et. al 2021; Chen et. al 2021]), and the resulting sample size requirement can be decreased to $\mathsf{poly}(s, \log d)$.
>
>
> > “I wonder if the $\ell_1/\ell_2$-norm values remain close to that of $p$ or not.”
>
> Thanks for bringing up this subtle but interesting point. Indeed, the $\ell_\infty$ form that we presented is slightly stronger than the standard $\ell_2$ (or $\ell_1$) guarantees. Observe that that the $\ell_\infty$ ball with radius $c/\sqrt{d}$ (denoted as $\mathcal{B}(\ell_\infty, \frac{c}{\sqrt{d}})$) is contained in $\mathcal{B}(\ell_2, c)$, so
> $$\sup_{ p’: \lVert p’ - p \rVert_\infty \leq \sqrt{\frac{\lVert p \rVert_{1/2}}{nd2^b}}} \mathbb{E}\lVert \hat{p} - p’ \rVert^2_2$$ is always greater than the supremum over
> $$\sup_{ p’: \lVert p’ - p \rVert_2 \leq \sqrt{\frac{\lVert p \rVert_{1/2}}{n2^b}} } \mathbb{E}\lVert \hat{p} - p’ \rVert^2_2.$$ A similar form can be obtained for the $\ell_1$ distance (in which we may use the fact that $\mathcal{B}(\ell_\infty, \frac{c}{\sqrt{d}}) \subseteq \mathcal{B}(\ell_1, c\sqrt{d})$).
>
> > “The optimality of the results on the $\ell_1$ norm is unclear.”
>
> We agree that the optimal $\ell_1$ convergence rate remains open, but we also want to emphasize that the $\ell_2$ error is equally important since it has been largely studied in the local minimax theory for both parametric or non-parametric estimation. On the other hand, we believe that combining our lower bound techniques with the recent result by [Sarbu and Zaidi 2021] (which generalizes the global minimax lower bounds in [Barnes et. al, 2019] to $\ell_p$ loss with $p \in (1, 2]$), we can obtain tight local minimax lower bounds for $\ell_p$ loss with $ p \in (1, 2]$. For $p=1$, we may have to use other information-theoretic tools (such as Assoud’s method [Acharya et. al, 2021]) with proper localization techniques. Bridging this gap is left as future work.
>
> > “ I can see the upper bound that shows the scheme is minimax optimal over $\mathcal{P}_{s, d}$, but not a (nearly) matching lower bound for $s$-sparse distributions.”
>
> The global minimax lower bound for $P_{s, d}$ is due to Theorem 2.3, not the lower bound from [10]. Note that the lower bound in [10] is derived under a non-interactive setting, while our lower bounds (both Theorem 2.3 and 2.4) are based on [Barnes et. al, 2019], which holds for general interactive settings. Thus our scheme is indeed minimax optimal over $\mathcal{P}_{s, d}$ under the sequentially interactive setting.
>
> > “Corollary 3.2 assumes only $\beta \in (0,1/3]$.”
>
> Thanks for catching it, this is indeed a typo, and the correct statement is “This result shows that $\textbf{if}$ $\beta$ is constant for the truncated geometric distribution, 1 bit suffices to achieve the centralized convergence rate”.
>
> > “Distributions with small effective support size may be learnable.”
>
> On the one hand, we would like to emphasize that [10] does not generalize to estimating near-sparse distributions. Even if we extend their analysis to the nearly sparse setting (or using other sparse estimation techniques), the net error would involve two terms: one for the approximation error (which is the error of using an $s$-sparse $p_s$ to approximate the true target $p$) and the other for the estimation error (which is the error of estimating $p_s$). Although the estimation error can be well-controlled and decreases with the sample size $n$, the approximation usually does not decay with $n$, making the resulting scheme strictly sub-optimal compared with our proposed one. See, for instance, Theorem 4.2 in [Xiong et. al, 2020] under the local differential privacy constraint. Moreover, methods based on the sparse estimation require prior knowledge on the sparsity $s$, while our method does not need this information.
>
> References:
> [Han et. al, 2018a] "Distributed statistical estimation of high dimensional and nonparametric distributions", ISIT 2018
> [Han et. al, 2018b] "Geometric lower bounds for distributed parameter estimation under communication constraints", COLT 2018
> [Barnes et. al, 2019] "Lower bounds for learning distributions under communication constraints via fisher information", JMLR 2020
> [Acharya et. al, 2021] "Estimating Sparse Discrete Distributions Under Privacy and Communication Constraints", ALT 2021
> [Chen et. al, 2021] "Breaking The Dimension Dependence in Sparse Distribution Estimation under Communication Constraints", COLT 2021
> [Sarbu and Zaidi, 2021] "On Learning Parametric Distributions from Quantized Samples", ISIT 2021
> [Xiong et. al, 2020]  "Compressive Privatization: Sparse Distribution Estimation under Locally Differential Privacy", arXiv 2012.02081

---

### Official Review · Reviewer_PqZC · 2021-07-22

**Rating:** 6
**Confidence:** 4

**Summary:**

This paper considers "instance-optimality" of discrete distribution estimation under communication constraints for interactive schemes where each client receives a single sample from the distribution p. The focus is on the small alphabet regime (d<<n). Upper bounds for $l_2$ error feature the half norm and for $l_1$ error feature the one-third norm.

**Limitations And Societal Impact:**

I am satisfied with the limitations that have been addressed.

**Main Review:**

The paper considers a problem of significant interest. This is the first work to consider instance-optimality in communication constrained settings. The writing is clear for most of the part, except the lower bound. It appears as if the authors copy-pasted parts of the lower bound proof in haste since there are symbols such as $S'_{\theta}(X)$ that have not been defined in the main text and there are references to equation numbers (7) and Lemma E.1 that can be found in the appendix.
Couple of concerns:
1) In the local refinement round, the knowledge of the groups is available at the server after the localization round. It's then not clear how this is available to clients since the knowledge of $\mathcal{J}_i$ is necessary to send the ranking of the symbol it receives.

2) Line 263 states that symbols with larger $p_j$ have a larger estimation error. It's not clear why this should be the case when one would expect that a higher probability symbol might be seen often and therefore be estimated more accurately.

Overall, I would have preferred more discussion about the intuition behind the local refinement scheme (for e.g. some explanation for why it works well). It is also worth pointing out in the related works section that most works that analyze the minimax settings focus on the large alphabet regime.

Some typos:
1) Pg 7, line 257, is the union an intersection?
2) Pg 8, line 263 - comma instead of period


**Time Spent Reviewing:**

4

---

> ### Author Response · Authors · 2021-08-10
> **Response to Reviewer PqZC**
>
> We thank the reviewer for the valuable comments and suggestions.
>
> > “It appears as if the authors copy-pasted parts of the lower bound proof in haste.”
>
> $S_\theta’(X)$ is the score function of X, formally defined as $\nabla_\theta\log(p_\theta(X))$. We will reorganize the proof in the revision and make sure the proof is clear and consistent.
>
>
> > “It's then not clear how this is available to clients.”
>
> We agree that the knowledge of $\mathcal{J}$  is necessary for the local refinement step; therefore we consider a (two-round) interactive setting (which is a canonical communication model that has been extensively considered in distributed learning/estimation literature), where all the previous transmitted messages $Y_1,...,Y_{i-1}$ are available at client $i$, and hence client $i$ can compute $\hat{p}$  (and thus $\mathcal{J}_i$). In practice, this can be done through the coordination of the server: after the localization step, the server computes $\hat{p}$ and broadcasts it to all clients participating in the local refinement step, and these clients can compute the corresponding $\mathcal{J}_i$ accordingly.
>
>
> > “It's not clear why symbols with larger $p_j$ have a larger estimation error.”
>
> In the localization step with uniform grouping strategy, each symbol $j$ is observed $n’ = n2^b/d$ times, so equivalently the server obtains $\mathsf{Binom}(n’, p_j)$ and can estimate $p_j$ by $\frac{1}{n’}\mathsf{Binom}(n’, p_j)$. Therefore the corresponding variance will be $\frac{p_j(1-p_j)}{n’}$, so larger $p_j$ (assuming $p_j \leq 1/2 $ ) results in larger estimation error.
>
> > “More discussion about the intuition behind the local refinement scheme.”
>
> As pointed out by Reviewer 3 (d6xD), the main intuition is that according to line 267 in our paper, the net $\ell_2$ error of the grouping scheme scales as $\sum_j p_j/n_j$, so optimizing over $n_j$ implies $n_j \propto \sqrt{p_j}$. Therefore we design our refinement scheme via the non-uniform grouping strategy with this particular choice of grouping size. We will elaborate more on the intuition behind the refinement step in the revision.
>
>
> > “It is also worth pointing out in the related works section that most works that analyze the minimax settings focus on the large alphabet regime.”
>
> We will clarify this point in the revision. However, we would also like to point out that even in the minimax settings, all of the previous works require the sample size to be larger than $\mathsf{poly}(d)$. For instance, the upper bound in [Han et. al, 2018a] requires $n \succeq d^2$, and the minimax lower bounds in [Han et. al, 2018b] and [Barnes et. al, 2019] also require $n \succeq d^2$.
>
> In addition, there are many applications where the number of samples is larger than the domain size, for example, emoji prediction task, hit song prediction, or sensor networks, etc. In some of these scenarios (such as sensor networks), the communication cost may be extremely stringent as each distributed sensor has limited power to transmit the signal it observes, and hence reducing the communication is crucial and of practical interest.
>
> We would also like to remark that with additional prior knowledge on the target distribution such as sparse/near-sparse assumptions, we can easily improve our two-stage scheme by replacing the uniform grouping localization step with other sparse estimation schemes (such as [Acharya et. al 2021; Chen et. al 2021]), and the resulting sample size requirement can be decreased to $\mathsf{poly}(s, \log d)$.
>
>
> > “Some typos”
>
> Thanks for catching the typos, we will correct them in the revision. Regarding the first one, it should be an intersection instead of a union.
>
> References:
> [Han et. al, 2018a] "Distributed statistical estimation of high dimensional and nonparametric distributions", ISIT 2018
> [Han et. al, 2018b] "Geometric lower bounds for distributed parameter estimation under communication constraints", COLT 2018
> [Barnes et. al, 2019] "Lower bounds for learning distributions under communication constraints via fisher information", JMLR 2020
> [Acharya et. al, 2021] "Estimating Sparse Discrete Distributions Under Privacy and Communication Constraints", ALT 2021
> [Chen et. al, 2021] "Breaking The Dimension Dependence in Sparse Distribution Estimation under Communication Constraints", COLT 2021

---

### Decision · Program_Chairs · 2021-09-27

**Decision:**

Accept (Poster)

**Comment:**

Based on the reviews and discussion, I am inclined to recommend the acceptance of this paper.  The reviewers generally appreciate the importance of the problem, and the contributions made, particularly studying instance-optimality in a communication-constrained setting.  However, the reviewers do maintain some concerns, some of which are quoted as follows:
- "I still think $n\ge d^3$ is restrictive (also much worse than $d^2$ you mentioned). Even in applications where the number of samples is larger than the domain size, $d^3$ easily blows up. As other quantitative results are available, it is important to include proper discussions such as why you believe the transition happens at $d^3$."
- "I don't agree with "the scaling of our lower can be made arbitrarly close to the upper bound" due to the extra vanishing $c_\delta$. I do appreciate the lower bound $\Vert p\Vert_{1/2} /\log d $ though."
- "Perhaps the biggest weakness of the analysis is the large error terms, which require a large number of clients to be effective. Equivalently, the results of the paper do not offer improvements on sample complexity of the tasks for a given bitrate."
- "the authors claimed that their min-max Lower Bound is stronger because they maximize over a larger domain, not making intuitive sense. For another example, the authors commented that the $\ell_2$ error is equally important as the $\ell_1$ error for assessing discrete distribution learners. I don't think this is the case in the literature, and the $\ell_2$-type results are often much easier to derive. In particular, the authors also admitted that the $\ell_1$ problem remains "open." "

I believe that these are valid concerns, but at the same time, none of them were viewed as strong reasons for rejection, and one reviewer was willing to champion the paper more strongly, hence my recommendation.

In the final version of the paper, I ask that the authors carefully take the above comments *and* all of the reviews into careful consideration.